# Representation Similarity Reveals Implicit Layer Grouping in Neural Networks

## Abstract

Providing human-understandable insights into the inner workings of neural networks is an important step toward achieving more explainable and trustworthy AI. Analyzing representations across neural layers has become a widely used approach for this purpose in various applications. In this work, we take a step toward a holistic understanding of neural layers by investigating the existence of distinct layer groupings within them. Specifically, we explore using representation similarity within neural networks to identify clusters of similar layers, revealing potential layer groupings. We achieve this by proposing, for the first time to our knowledge, the use of Gromov-Wasserstein distance, which overcomes challenges posed by varying distributions and dimensionalities across intermediate representations–issues that complicate direct layer-to-layer comparisons. On algebraic, language, and vision tasks, we observe the emergence of layer groups that correspond to functional abstractions within networks. These results reveal implicit layer structure pattern, and suggest that the network computations may exhibit abrupt shifts rather than smooth transitions. Through downstream applications of model compression and fine-tuning, we validate our measure and further show the proposed approach offers meaningful insights into the internal behavior of neural networks.

## 1 Introduction

Recent advances in neural networks, in particular large models, has prompted increased interest in understanding the underlying causes of new capabilities. Since neural models, including large language models (LLMs), are mostly black-box models, explainable AI aims to offer insights and improve human understanding of these neural models. A widely adopted strategy involves analyzing the inner representations across network layers, as these offer valuable information about how intermediate computations evolve. As a result, measures to quantify representation similarity of these layers have been widely applied in the literature to gain novel insights Klabunde et al. (2023), including learning dynamics, impact of different network hyperparameters, knowledge distillation, and more.

Despite substantial progress, relatively little attention has been devoted to understanding the structure of the layers themselves. Neural networks, especially those with many layers[1] and parameters, often exhibit behaviors that are difficult to interpret holistically. Given such depth and opacity of modern networks, identifying coherent layer groupings can serve as a principled way to decompose models into more interpretable sub-units. Such structural insights may illuminate how specific portions of a network contribute to its overall functionality and enable a more modular understanding of computation.

In this work, we ask a fundamental question: *can we detect the existence of distinct layer groups within a trained neural network?* We answer this in the affirmative by demonstrating that representation similarity across layers can reveal meaningful structural boundaries. Surprisingly, our results indicate that layer transitions are not always smooth; instead, networks may exhibit abrupt shifts in computation, manifesting as changes in representation.

---

[1]The term layer is used broadly to refer to any intermediate or final output produced by internal computations. The generality of the term allows it to encompass everything from input preprocessing to hidden states and final predictions.

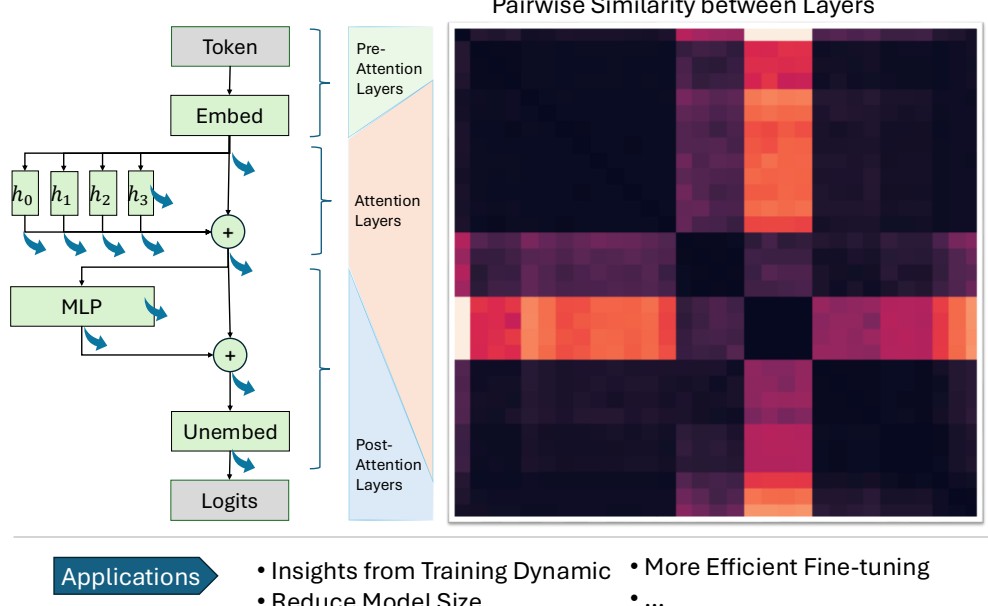

Figure 1: Overview of our approach, where we use representations from different neural network layers to identify functionally distinct layer groups (such as different darker-colored blocks) leveraging GW distance. The figure is an illustration based on a single transformer block, and the proposed technique can be applied to other types of network structures.

Comparing representation across different layers is natural, as it is akin to comparing functions by evaluating their outputs on shared inputs. This leads to the task of identifying layer groups as regions of functional similarity, with significant shifts marking transitions between layer groups. As illustrated in Figure 1, representational-similar layers (darker colors) cluster together, while less similar layers (brighter colors) highlight potential boundaries between groups.

A central challenge in this endeavor is defining a robust similarity measure. Standard metrics (e.g., cosine, Euclidean) are inadequate due to the varying dimensionality and metric space of layer outputs, especially in architectures like CNNs and transformers. Additionally, representations should be invariant to transformations such as rotation, scaling, permutation, and reflection. While many specialized similarity measures have been proposed, it is not clear which measure would be better for measuring the representation similarity. To this end, we propose to measure representation similarity using Gromov-Wasserstein (GW) distance (Zheng et al., 2022) between representations from different layers of the network. As elaborated further in Section 4.1, GW allows distance computation between distributions supported on two different metric spaces with different supports and potentially different dimensions, which is common across different layers in neural networks. GW is also invariant to permutation of the representation within a layer, a crucial property since neural networks are known to have permutation symmetries (Goodfellow et al., 2016). As such, GW can effectively identify genuinely distinct behaviors across (groups of) layers.

We validate our approach on algebraic, language, and vision tasks, showing that GW distance provides a systematic and architecture-agnostic way to analyze and identify layer structure. Additionally, our findings provides a holistic view on differences in representations of models trained with different strategies. We observe clear patterns in the form of block structures among different layers, suggesting there exist layer group that potentially compute different functions, particularly at the transition layers where major functional changes may occur. Moreover, our approach is applicable to various downstream tasks, such as tracking the emergence of layer groups during training process (§ 5.3) and identifying potentially redundant layers in model compression and fine-tuning (§ 5.2) Overall, our method can improve the efficiency of mechanistic interpretation by finding layer groups in any neural model, reducing the need for extensive human effort and contributing to a further understanding of neural network behaviors.

## 2 Background and Related Work

Neural network interpretability has always been a hot research topic, and has become a disparate area with many different applications in vision (Palit et al., 2023), and language (Ortu et al., 2024; Hernandez et al., 2023; Yu et al., 2024), and we survey a few closely related directions within it.

**Layer Grouping v.s. Other Types of Subnetwork** Automated circuit discovery (Conmy et al., 2024; Shi et al., 2024) aims to find a computational graph that is much more sparse without sacrificing performance. Moreover, Various work have explored neuron semantics and possible disentanglement of semantics (Bricken et al., 2023; Dreyer et al., 2024; Huang et al., 2024a) and concepts (Park et al., 2024). Instead of studying circuits and neurons, we investigate differences among neural network layers as a whole, based on an existing line of work (Nanda et al., 2023). Related, block structures within neural network layers have been observed in previous studies (Nguyen et al., 2022).

**Representation Similarity v.s. Other Methods of Studying Mechanisms** There are many other approach to understand neural network inner mechanism, such as weight inspection and manipulation (Ortu et al., 2024; Meng et al., 2022). Modifying or ablating the representation of a specific model components (Huang et al., 2024b; Kramár et al., 2024), including attention knockout (Wang et al., 2022), and even direct modification of attention matrix (Ortu et al., 2024; Geva et al., 2023) are prevalent. Another popular approach involves inspection of representation properties (Meng et al., 2022), including logit patterns (Zhong et al., 2024), residual stream (Ortu et al., 2024), and periodicity (Nanda et al., 2023). Rather than just inspection of representation, many works have proposed to map the output to some target and is a popular technique for analyzing how neural activations correlate with high-level concepts (Huang et al., 2024a; Hou et al., 2023). Here we instead focus on studying and compare representation similarity across layers.

**Similarity Measure between Neural Network Layers** There are studies that quantify the similarity between different groups of neurons (Klabunde et al., 2023), typically layers (Ding et al., 2021; Yu et al., 2024), to compare different neural networks. Generally a normalized representation with desired properties is used to compare different transformer blocks, such as invariance to invertible linear transformation in canonical correlation analysis (Morcos et al., 2018), orthogonal transformation, isotropic scaling, and different initializations in centered kernel alignment (Kornblith et al., 2019). Other measures include representational similarity analysis (Mehrer et al., 2020), which studies all pairwise distances across different inputs. Wasserstein distance has been explored in measuring similarities in the context of neural networks (Dwivedi & Roig, 2019; Cao et al., 2022; Lohit & Jones, 2022), but they assume that different layer representations belong to the same metric space, which is very unlikely even if they have the same dimensionality as the semantics captured by each layer are likely to differ significantly. Several similarity measures (Tsitsulin et al., 2019; Demetci et al., 2023a) seek different approximation or or addition to GW distance. While GW distance has been used for model merging as a regularization (Singh & Jaggi, 2020; Stoica et al., 2023), it has not been fully explored in discovering layer structures and subnetwork identification.

## 3 Representation Similarity Within Neural Networks

We aim to identify layer groupings of a neural network based on their representation at each layer. To find these layer groups, the problem can be formulated as a search problem: among candidate layers, we seek to find which candidate layers are the most dissimilar to its previous layers, indicating the formation of a new group.

The problem consisting of two key elements: the search space and the similarity measures used to evaluate how closely the candidates in the search space match another (target) representation. We first discuss the notation of representation similarity as the measure for searching and the search space for candidates. We then discuss specific choices on the similarity measure.

**Similarity Measures** Let $f : X \to Y$ be a function that map $x$ in a set of input $X = \{x_i : x_i \in \mathbb{R}^{d_x}\}_{i=1}^n$ to $y$ in a set of output $Y = \{y_i : y_i \in \mathbb{R}^{d_y}\}_{i=1}^n$. Each element in $Y$ and $X$ are assumed to be a vector with dimensions $d_y$ and $d_x$, respectively, with $n$ being the set size, without loss of generality. Note sets can be concatenated into matrix forms as $Y \in \mathbb{R}^{n \times d_y}$ and $X \in \mathbb{R}^{n \times d_x}$.

**Definition 3.1. Representation Similarity**. Consider neural networks have the form $f = f^{(L)} \circ f^{(L-1)} \circ \cdots \circ f^{(1)}(X)$, where each layer $l \in \{1, \ldots, L\}$ computes a function $\mathcal{F}^{(l)}(X) = f^{(l)} \circ f^{(l-1)} \circ \cdots \circ f^{(1)}(X)$ given the input $X$. Representation similarity between two neural layers $i$ and $j$ can be seen as a similarity between their output sets $Y_i = \mathcal{F}^{(i)}(X)$ and $Y_j = \mathcal{F}^{(j)}(X)$, over the same set of input $X$.

Each intermediate representations of a neural network can be naturally treated as function outputs, given inputs $X$. We can use a scoring or distance function $D(Y^i, Y^j)$ as a measure between representations similarity between $\mathcal{F}^{(i)}(X)$ and $\mathcal{F}^{(j)}(X)$. If they are close according to $D$, then the layer should be similar to each other locally at a set of points $X$. Otherwise, these layers should be different at $X$. Popular measures such as Euclidean distance have been used for this purpose (Klabunde et al., 2023).

**The Need for Complex Representation Similarity Measure.** Since we cannot exactly control the behavior of a trained neural network, the layer-wise functions $\mathcal{F}$ that it learns can be complex and thus the learned representation $Y$ from each layer may be a complex function from each other rather than a simpler transformation. For example, let $Y_i = \sin(X)$ and $Y_j = \sin^2(X) = (Y^i)^2$. They share strong similarity, but a linear transformation will have trouble to fully capture their similarity. If we want to truly understand where function $\mathcal{F}$ might be approximately computed, we should consider some functions of target $Y$, but naively listing out all possibilities can be prohibitive. As a consequence, one may need to use more complex measures to deal with such a space.

**Search Space** We consider multiple candidate $Y$'s to form the search space for comparison. In the context of MLP neural networks for example, where $\sigma(.)$ denotes the non-linearity and $W$s are the parameter matrices, we have $Y^* = W_n(\sigma(W_{n-1} \ldots \sigma(W_1 X)))$ for the whole network. We can extract many $Y$'s from intermediate layers of the model, for instance $Y_1 = W_1 X$, $Y_2 = \sigma(W_1 X)$, and so on. These $Y$'s are often called representations, activations, or sometimes even "outputs" from each layer. We use these terms interchangeably here. For attention modules in transformer neural networks (Vaswani et al., 2017), we can similarly extract $Y$'s from attention key, query, and value functions as well as MLP functions. We list the exact equations and locations of representations considered in the transformer models in Table 4 in Appendix A, which serves as the focus of this paper. We consider attention-based models first, and later we also consider convolutional neural networks with residual layers, with candidate representations listed in Table 5 in in Appendix A.

## 4 Gromov-Wasserstein Distance as a Similarity Measure

We aim to identify the similarities among the representations at each intermediate layer. Each layer, however, posits a representation that potentially has a different distribution, not to mention even different dimensionality depending on the architectures and layers one considers (viz. mlp and attention layers in transformer blocks). Consequently, representations across layers may be incomparable using standard distance metrics, such as the $\ell_p$ norm amongst others.

To address these challenges, we propose computing distances between representations at the same layers for different inputs, and match the vertices of a weighted graph – where each dimension of the representation are vertices and the distances indicate weights on the edges – with the vertices of a similarly constructed weighted graph from another layer. Essentially, we assume the representations in a layer are samples of the underlying distribution, and we want the best permutation of representation dimensions in one layer that aligns with vertices in another layer, thereby deriving the inter-layer distance. If this inter-layer distance is low, then we consider the two layers similar.

Formally, without loss of generality, let $Y_1 = \{y_{1i} : y_{1i} \in \mathbb{R}^{d_1}\}_{i=1}^n$ and $Y_2 = \{y_{2i} : y_{2i} \in \mathbb{R}^{d_2}\}_{i=1}^n$ be representations of $n$ examples from two different layers, where the discrete distributions over the representations are $\mu_1$ and $\mu_2$ respectively, with dimension $d_1$ possibly being different from $d_2$. Direct distance computation between them is not reasonable. Instead, we seek to compute a coupling or matching $\pi \in \Pi(\mu_1, \mu_2)$ between the $n$ examples in each set such that given the pairwise distances $D_1, D_2 \in \mathbb{R}^{n \times n}$ within representations $\boldsymbol{Y}_1$ and $\boldsymbol{Y}_2$ respectively, the sum of differences between the distances of the matched examples is minimized. Loosely speaking, we aim to find a matching that preserves the pairwise distance as much as possible. In

particular, we want to minimize the following:

$$\rho(\boldsymbol{Y}_1, \boldsymbol{Y}_2, \mu_1, \mu_2, D_1, D_2) \triangleq$$
$$\min_{\pi \in \Pi(\mu_1, \mu_2)} \sum_{i,j,k,l} (D_1(i,k) - D_2(j,l))^2 \pi_{i,j} \pi_{k,l}$$
$$\text{s. t.} \quad \pi \boldsymbol{I} = \mu_1; \pi^T \boldsymbol{I} = \mu_2; \pi \geq 0. \tag{1}$$

It turns out that $\rho$ corresponds to the Gromov-Wasserstein (GW) distance (Demetci et al., 2023b), used to map two sets of points in optimal transport. We thus utilize this distance as a measure of inter-layer functional similarity in the setting where the target is unknown.

### 4.1 Justification for GW Distance as a Functional Similarity Measure

Let $(\boldsymbol{Y}_1, D_1, \mu_1)$ and $(\boldsymbol{Y}_2, D_2, \mu_2)$ be two given metric measure space (mm-space), where $(\boldsymbol{Y}, D)$ is a compact metric space and $\mu$ is a Borel probability measure with full support: $\text{supp}(\mu) = \boldsymbol{Y}$. An isomorphism between $\boldsymbol{Y}_1, \boldsymbol{Y}_2$ is any isometry $\Psi : \boldsymbol{Y}_1 \rightarrow \boldsymbol{Y}_2$, i.e., a distance-preserving transformation between metric spaces, such that $\Psi_{\#\mu_1} = \mu(\Psi^{-1}) = \mu_2$.

**Theorem 4.1.** *(Mémoli, 2011). The Gromov-Wasserstein distance in equation 1 defines a proper distance on the collection of isomorphism classes of the mm-spaces.*

*Remark.* The Gromov-Wasserstein distance itself is defined on isomorphism-classes of metric measure spaces, which means that any distance preserving (isometric) transformation of a space should preserve GW distance between the points in that space and any other space (Mémoli, 2011). These isometric transformations include rigid motions (translations and rotations) and reflections or compositions of them. Additionally, permutations of points in a space also preserve GW distances, as the points are unlabeled.

In comparison, the GW distance is not invariant to several transformations that do not preserve the original pairwise distances or metric structure of the spaces being compared. Some examples include: 1) Scaling: multiplying coordinates by a constant changes all pairwise distances, so GW changes, 2) Shearing: a type of linear transformation that shifts points parallel to a fixed line (or plane) or fixed direction, by an amount proportional to their distance from that line, which distorts angles and relative distances. 3) Non-isometric affine transformations: examples include stretching, compressing, or skewing the space along specific axes. Lastly, 4) Nonlinear distortions: any transformation that non-uniformly warps or deforms the space in a way that is not an affine transformation would change the GW distance.

The computed GW distance represents the minimal distance over all possible transportation plans between two sets of points from different spaces. Since GW can be viewed as a measure that quantifies the distance between distance-based (i.g., Euclidean-distance) graphs, with a set of points as its nodes, GW distance would be low if the graph undergoes (nearly) isomorphic transformations between layers. Conversely, a high GW distance indicates a non-distance preserving transformation across layers, potentially reflecting a complex operation. While GW distance does not reveal the exact function operation, it highlights specific layers for further investigations.

**Favorable Properties of GW.** Besides the above noteworthy property of GW, it also has other favorable properties (Zheng et al., 2022; Demetci et al., 2023b): i) It is symmetric and satisfies the triangle inequality. ii) It is invariant under any isometric transformation of the input, which is advantageous because we do not want rotations and reflections to affect our similarity search. This invariance also includes permutation invariance, which is beneficial since the distance between layer representations should remain unaffected by permutations of neurons. iii) GW is scalable since it does not require estimating high-dimensional distributions, which is scalable to larger hidden dimensions of intermediate layers; instead, it only compare them to obtain a distance measure. iv) GW is monotonic in (positive) scaling of pairwise distances, and hence similar layers should appear to be closer than others with scaling.

**Distance Distributions.** As an illustrative example, we plot the histogram on pairwise distances for a batch of samples across all transformer blocks in BERT models from the YELP review dataset in Figure 2a. For more details on YELP, we provide a comprehensive discussion of experiments § 5.2. The results in Figure 2a

show the distributions on pairwise distances begin to differ from block 9, consistent with GW distance observed in Figure 5, suggesting that significant transformations occur and can be effectively captured by GW. We include the full results and discussion in Appendix E.

**Neighborhood Change.** Complementary to the distribution of pairwise distances, the changing representations of samples could also alter their relative neighborhoods across transformer blocks. We plot a tSNE projection (Van der Maaten & Hinton, 2008) of representations from a batch of samples on YELP, and visualize it in Figure 2b. The Jaccard similarity, measuring the overlap between top-5-neighbors of 3 selected samples across different transformer blocks, ranges from 0.0 to 0.43, with average values of $\{0.27, 0.26, 0.26\}$. The full details are shown in Table 7 of Appendix E. Hence, the sample neighborhood changes across blocks, which can be indicative of functional changes that are not captured by comparing distributions alone. However, GW can account for such changes as well.

**Computation Details.** We use an existing optimal transport toolbox, `pythonot` (Flamary et al., 2021), for computing GW distance. Specifically, we use an approximate conditional gradient algorithm proposed in (Titouan et al., 2019), which has a complexity of $O(mn^2 + m^2n)$, where $m$ and $n$ are the dimensions of two spaces (here the number of data samples from two layers being compared). In comparison, the Wasserstein distance Lohit & Jones (2022) may require $O(n^3log(n))$ for exact computation. When the dataset is large, we can also sub-sample the dataset to improve the computational efficiency.

# 5 Empirical Study and Findings

We compare the proposed similarity measure for layer grouping against a set of baselines across multiple datasets, including those from algebraic operation, NLP, and computer vision tasks. We compare 10 different similarity measures, including Euclidean, cosine, mutual information (MI), RSM(Klabunde et al., 2023), RSA(Klabunde et al., 2023), CCA(Morcos et al., 2018), CKA(Kornblith et al., 2019), MSIDTsitsulin et al. (2019), Wasserstein(Dwivedi & Roig, 2019), and AGW(Demetci et al., 2023a). For more details about implementation, please see Appendix F. All experiments are done on a single machine with 3.2 GHz CPU and 64 GB memory.

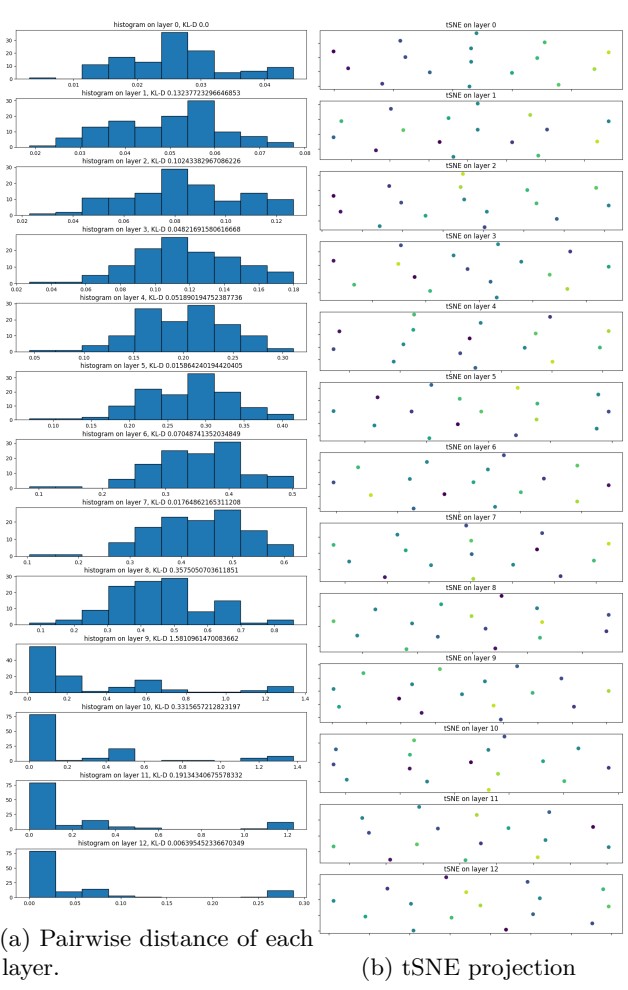

(a) Pairwise distance of each layer.

(b) tSNE projection

Figure 2: a) Histogram on pairwise distances for outputs from all transformer blocks in a fine-tuned BERT model trained on YELP dataset. a) layer 0 to layer 12. *b*) shows the neighborhood of tSNE projection for a batch of data across different layers.

## 5.1 Validation: Synthetic Modular Sum Tasks

Modular sum algorithms (Nanda et al., 2023; Zhong et al., 2024) and related math problems (He et al., 2024; Charton, 2023) are often studied to understanding the NN computation due to their precise function definitions. These works have shown that there may exist many different sub-functions in the computation, at different part of neural network. We begin by validating the Gromov-Wasserstein distance by comparing it against

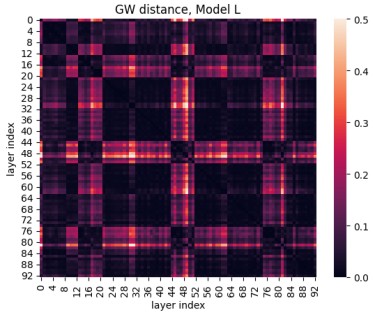
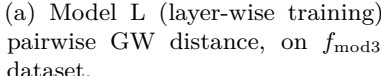

(a) Model L (layer-wise training) pairwise GW distance, on $f_{\mathrm{mod3}}$ dataset.

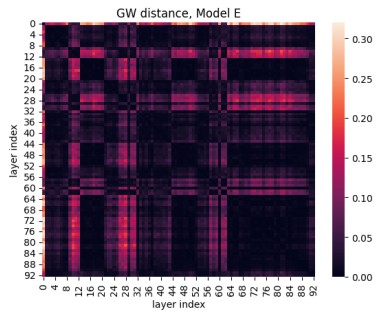
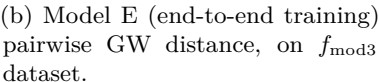

(b) Model E (end-to-end training) pairwise GW distance, on $f_{\mathrm{mod3}}$ dataset.

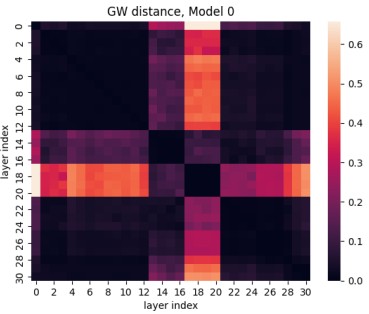

(c) Model 0 Pairwise GW distance, on $f_{\mathrm{mod}}$ dataset.

Figure 3: Pairwise GW distance on the synthetic Modular Sum dataset.

known partitions of the networks to determine whether it can successfully identify different layer groups. We first introduce the setup for the experiment, including data generation and models to be investigated.

**Setup** As a test case, we focus on a modular sum problem, following existing works (Nanda et al., 2023). We consider two datasets: the first generated by a single modular sum function with $c = f_{\mathrm{mod}}(a+b) = (a+b)\mathrm{mod}\ \mathrm{p}$, where $a, b, c = 0, 1, \ldots, p-1$, with $p = 59$. The second dataset is more complex, with $c = f_{\mathrm{mod3}}(a, b)$ of three levels of modular sums, namely: $c_1 = (a+b)\mathrm{mod}\ p_1, c_2 = (c_1+b)\mathrm{mod}\ p_2, c = (c_2+b)\mathrm{mod}\ p_3$, where $p = [59, 31, 17]$.

**Training procedure** We train 3 different neural networks with transformer blocks to predict $c$ given $(a, b)$, with learned input embedding of size $d$ and a decoding layer for categorical output. For the first simpler $f_{\mathrm{mod}}$ dataset, we train a neural network consisting of a one-block ReLU transformer (Vaswani et al., 2017), following the same protocol and hyperparameter choices as previous works (Nanda et al., 2023; Zhong et al., 2024). We call this **Model 0**. For the more complex $f_{\mathrm{mod3}}$ dataset, we train two neural networks consisting of three-block ReLU transformers, with 3 transformer blocks corresponding to the three levels of modular sum functions, and 4 attention heads within each block. The first network, which we call **Model E**, employs an end-to-end training procedure to directly learn output $c$. For the second network, which we call **Model L**, we use the same architecture as Model E but is trained layer-wise: block 1 predicts $(c_1, b)$, block 2 predicts $(c_2, b)$ with frozen earlier layers, and block 3 predicts $c$ from $(c_2, b)$. All models achieve 100% accuracy on a held-out validation set. More details can be found in appendix B.

Table 1: Gromov-Wasserstein Distance Results for Various Targets in Model L, for $f_{\mathrm{mod3}}$ dataset.

| GW-D for | Top Similar Layers | $D_{\min} =$ |
|---|---|---|
| $c_1$ | Resid-Post[1] | 0.02 |
| $c_2$ | Resid-Post[2] | 0.03 |
| $c$ | Resid-Post[2], Resid-Post[3], and 7 others | 0.04 |

**Results** The results are shown in the Table 1. We see that in the **Model L**, the GW distance correctly identifies the most similar layers in accordance with different intermediate $c$'s. The final target $c$ contains 9 similar layers all with distance around 0.04. In Appendix C, we also test probes as the prediction targets are given. Results shows GW distance can be a reliable alternative to the probes to find layer structures. Moreover, as previously mentioned GW distance can naturally compare representations across and within transformer blocks with different dimensions. In Fig 3a and Fig 3b, we visualize the pairwise GW distance between layer representations without a target for **Model L** and **Model E**. Looking at **Model L** we see predominantly 3 groupings of layers: i) layers roughly from 20 to 44 are similar to each other and to layers 52 to 72, ii) layers roughly 12 to 19 are similar to each other and layers 45 to 51 and iii) the initial and last few layers are mainly similar to themselves. Interestingly, the number of groupings corresponds to the 3 functions trained layer-wise in **Model L**. We also observe differences in patterns across **Model L** and **Model E**, suggesting layer-wise and end-to-end training return different networks. Compared to the fixed layer-wise training, end-to-end training in **Model E** may learn faster in the earlier layers and may not have much

to learn in later layers, as the function may not be particularly challenging for it. This could explain why, starting from layer 64, all layers in **Model E** exhibit similar representations. Moreover, magnitudes of the distances are also different, with **Model L** showing larger distances, indicating that learning the targets $c_1, c_2$ result in more functional differences. One possible explanation could be that **Model E** directly operates in the trigonometry space (Nanda et al., 2023), without having to predict the exact integer values until later, thereby suppressing the distances.

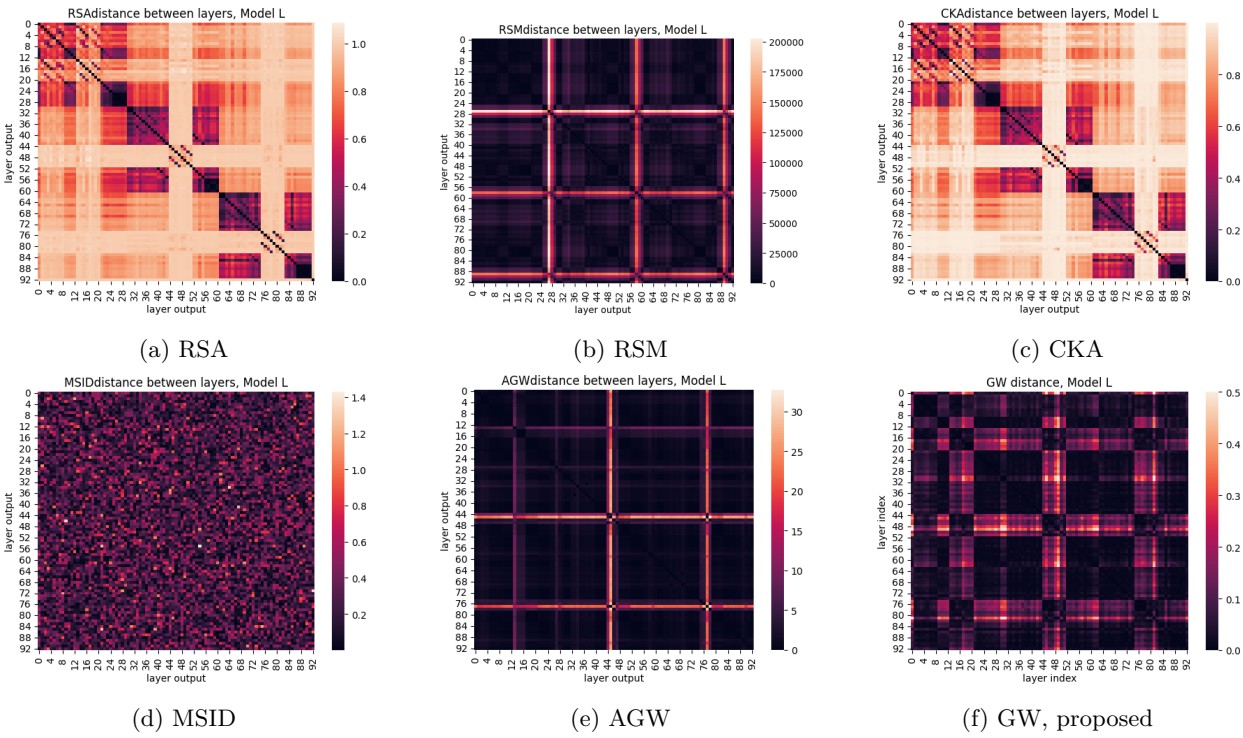

Figure 4: Pairwise (layer) distances on Modular Sum dataset, with layer-wise trained models. Different figures from left to right, top to bottom: *RSA, RSM, CKA, MSID, AGW, and the proposed GW distance.*

To gain a deeper understanding of the operations within each transformer block, we visualize pairwise GW distances among layers for **Model 0** for dataset $f_{\mathrm{mod}}$ in Figure 3c. In this case, we have a total of 31 representations since only one transformer block is used. We notice the first major difference occurs between layers 13 and 16, which are 4 Attn-Pre (computing key and value product). The second difference occurs between layers 17 and 20, which are the first 3 Attn (computing $A(X)$). This suggests that major computation seems to be done by the attention mechanism. Note that distances are not monotonically increasing across layers, which is expected as the representation spaces can change significantly given the heterogeneity of the operations such as those performed by residual connections and attention within a transformer block.

We have also tested a few baselines that can handle different space dimensions, shown in Figure 4. RSA and CKA reveal different levels of lay grouping within attention layers and across transformer blocks. AGW demonstrates the highest sensitivity to attention computations, while RSM finds the last few layers within each transformer block.

## 5.2 Real Dataset: NLP Tasks

**Setup** We now apply GW distance to real natural language processing tasks. We experiment on benchmark sentiment analysis datasets, Yelp reviews and Stanford Sentiment Treebank-v2 (SST2) from the GLUE NLP benchmark (Wang et al., 2019), with the goal to predict of the text has positive or negative sentiment, and analyze how different layers from fine-tuning BERT(-base) (Devlin et al., 2019) models perform on these datasets. We use the pretrained BERT to generate 4 fine-tuned models, corresponding to a dense model and 3 sparse models with sparsity levels of 25%, 70% and 95% using a state-of-the-art structured pruning

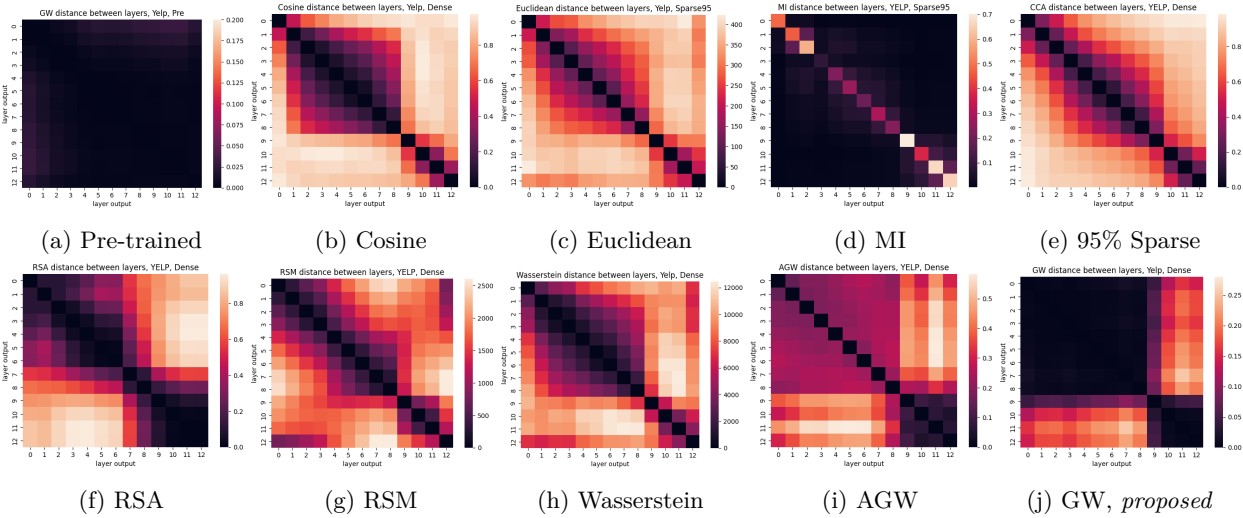

Figure 5: Pairwise (layer) distances on Yelp, across pre-train and densely fine-tuned BERT models, using the various similarity measure. As can be seen GW clearly demarcates the layer groups. Due to page limit, we show more results in Appendix G.

algorithm (Dhurandhar et al., 2024), but only show results on densely fine-tuned models here. Training details and more results are in Appendix D and G. Due to the size of BERT models, we limit our analysis to comparing the final representations from each of the 12 transformer blocks, rather than examining all intermediate representations within transformer blocks.

**Results** In Figure 5a, we see that the pre-trained BERT does not have major differences among blocks, which is not surprising given its accuracy on YELP is only 49.3% (roughly equivalent to random guessing). In Figures 5b to 5j, we show similarity measures from difference baselines. We can see an interesting pattern emerge, revealing two major block structures in the fine-tuned BERT models identified by our approach. The first major differences occur at block 9 and then the last three blocks (10, 11, 12) seem to form a distinct block. This seems to indicate that most of the function/task fitting occurs at these later blocks. GW distance gives the most distinctive layer groups.

The pattern indicates that later blocks differ significantly from earlier ones. This observation is consistent with fine-tuning and sparsification literature (Li et al., 2021; Dhurandhar et al., 2024), where it has been observed that later blocks typically undergo substantial changes during fine-tuning as they focus on task-specific solutions, while the earlier middle blocks remain stable as they capture syntactic and semantic patterns of the language necessary for various tasks. In Appendix H, we further investigate the GW distance between blocks from two different models, providing insights into how representations vary across architectures. In Appendix J, we vary the seeds for training and observe consistent patterns over different seeds. In Appendix G, we include results from baseline similarity measures. Overall, CKA produces also similar block structures to the proposed GW distance, though with greater variability within block structures. In contrast, other baselines fail to reveal such clear block structures.

On SST2 dataset, we also observe very similar patterns with the GW distance and 3 baselines, for which we refer the readers to appendix I for detailed results. In both datasets, low distance measure are consistently observed in the diagonal elements, but the overall block structures are not as obvious in the baselines as they are with GW distance, highlighting the effectiveness of the GW distance.

**Model Compression** The presence of block structures in the GW-distance matrices indicates major functional changes may concentrate at these transition blocks. This finding may suggest that for other downstream tasks, we may consider freezing the model up to Block 8 and only fine-tuning the blocks after that. Since the GW distance indicates significant changes occurs only at later layers in YELPS, we investigate performance of fine-tuning only partial layers from pretrained models, by freezing early layers during training

and training only later layers alongside a classification layer (denoted as C) at the end. In Table 2, we can see that there is no significant performance differences between fine-tuning layer 8 to 12 and fine-tuning layer 9 to 12 (0.04% drop). On the other hand, the accuracy drops 6 times more by freezing layer 1 to 9, with 0.25%. Freezing layer 1 to 10 results 0.49% drop, and finally fine-tuning only 12 results 3.59% drop. These findings validate that the later layers are crucial for significant functional changes.

Table 2: Accuracy of fine-tuning partial layers in various BERT models. C denotes the classification layer on top of BERT models.

| Fine-tune | All | 8~12 + C | 9~12 + C | 10~12 + C | 11~12 + C | Only 12 + C |
|---|---|---|---|---|---|---|
| Accuracy (%) | 97.87 | 97.47 | 97.43 | 97.19 | 96.7 | 93.11 |

**Model Pruning** Another another potential application beside freezing-and-fine-tuning specific transformer blocks, we study the problem of model compress or pruning with the discovered layer groupings.

For each of desired block sizes, we take the original pre-trained BERT and only use the first $n = \{12, 8, 4, 2, 1, 0\}$ transformer blocks while discarding the rest. Note that $n = 12$ means we use all the transformer blocks, resulting the same BERT model. $n = 0$, on the other hand, means that we only use a (linear) classifier layer (after embedding layer) to predict the class label. The results are shown in Table 3. As a reminder, GW distance suggest the last 4 blocks in YELP (see Figure 5) and the last 2 blocks in SST (see Figure 15) are mostly different, which is marked by star ($*$) in the table. It shows that by using a limited number of layers, we can achieve similar performance with the full 12 block model, with 0.01% and 0.54% differences in YELP and SST, respectively. Using one fewer transformer block can risk much worse reduction of performance, with 0.10% and 8.60% differences (about 10 times worse performance reduction).

Table 3: Accuracy of pruning BERT with a smaller number of blocks on YELP and SST. N denotes the number of transformer blocks in the new BERT models.

| Number of Transformer Blocks | 12 (all) | 8 | 4 | 2 | 1 | 0 (only classifier) |
|---|---|---|---|---|---|---|
| YELP | 97.87 | 97.87 | 97.86* | 97.76 | 97.11 | 60.3 |
| SST | 92.40 | 90.25 | 90.25 | 91.86* | 83.26 | 50.92 |

**Clustering** Besides visualization above, one can also utilize clustering methods to automatically identify the layer groups from the GW distance. We tested spectral clustering (Von Luxburg, 2007) on a similarity matrix computed as the reverse pairwise GW distance matrix. This method successfully identified 2 groups with block $1 \sim 8$ and block $9 \sim 12$. Note that one could use a clustering algorithm and use some quantitative evaluations, such as clustering metrics like silhouette scores, to measure which metrics provide better clusters. However, since different metrics produce different scores, it is hard to compare clustering groups when the inputs to a cluster algorithm are different.

### 5.3 Training Dynamics: Emergence of Layer Groups During Training

We also visualize the GW distance between blocks while fine-tuning the pretrained BERT model on YELP datasets in the entire training process, in order to observe when these layer grouping structures begin to emerge. Figure 6 show a few visualization on GW distances at selected training iterations. Block distances are low in the beginning (observed in Figure 5a), but by iteration 300 the last block begin to differ from other blocks. As training progresses, block 9, 10, and 11 begin to show at iteration 3k and 15k. These growing differences in GW distance reflect the model's increasing F1 score on the test data. Overall it show the gradual specialization of blocks into distinct sub-networks, with each sub-network potentially focusing on different aspects of the task.

We also plot the mean GW distance of all block pairs in Figure 7. Figure 7a show the mean GW distance over training iterations, and show it grows over time. Figure 7b shows that mean GW distance versus two different accuracy metric on the test dataset. GW distance grow slowly at first, followed by a rapid increase

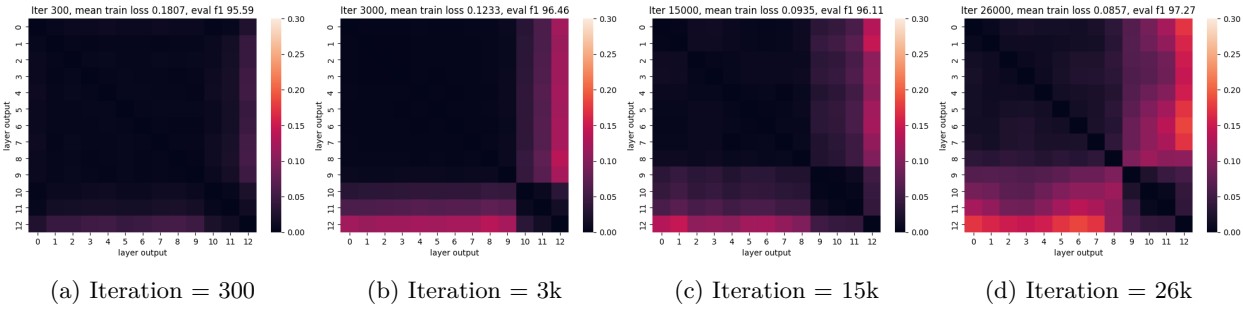

| (a) Iteration = 300 | (b) Iteration = 3k | (c) Iteration = 15k | (d) Iteration = 26k |

Figure 6: Pairwise GW distance in YELP datasets, over training iterations.

as the model achieves better accuracy and F1 scores. Such observation is consistent with existing "grokking" behavior, where validation accuracy can suddenly increases well after achieving near perfect training accuracy (Nanda et al., 2023). Similarly, Figure 7c shows a rapid increase in mean GW distance in order to achieve a lower training loss.

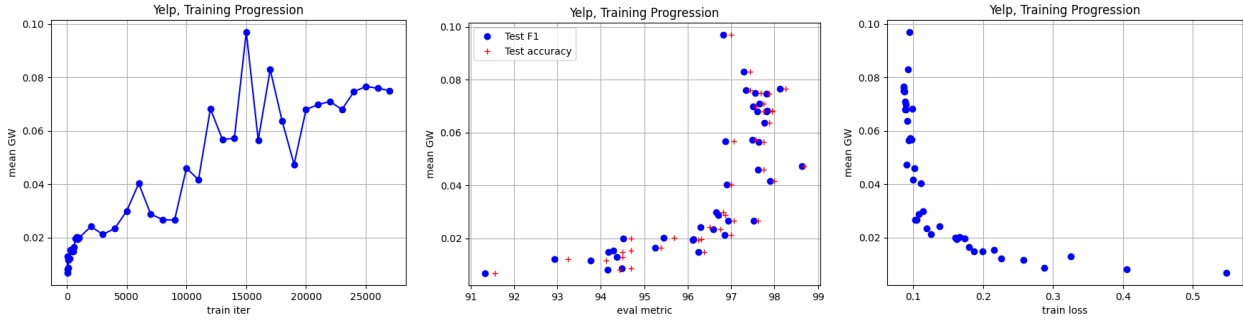

Figure 7: Pairwise GW distance in YELP datasets, over training iterations.

## 5.4 ResNet and Computer Vision Dataset

In addition to the attention-based architectures, we also test our approach on ResNet9, a popular convolutional neural network architecture (He et al., 2016; Park et al., 2023). We compare a randomly initialized ResNet9 and a trained one on CIFAR 10 image dataset CIFAR-10 (Krizhevsky et al., 2009), achieving 91.63% accuracy on the test data. For more details on the setup, we refer the readers to Appendix K. In Figure 18 of Appendix K, we show the pairwise distance among layers using baselines that handle different input dimensions. Overall, GW distance show the most clear divisions of layer grouping.

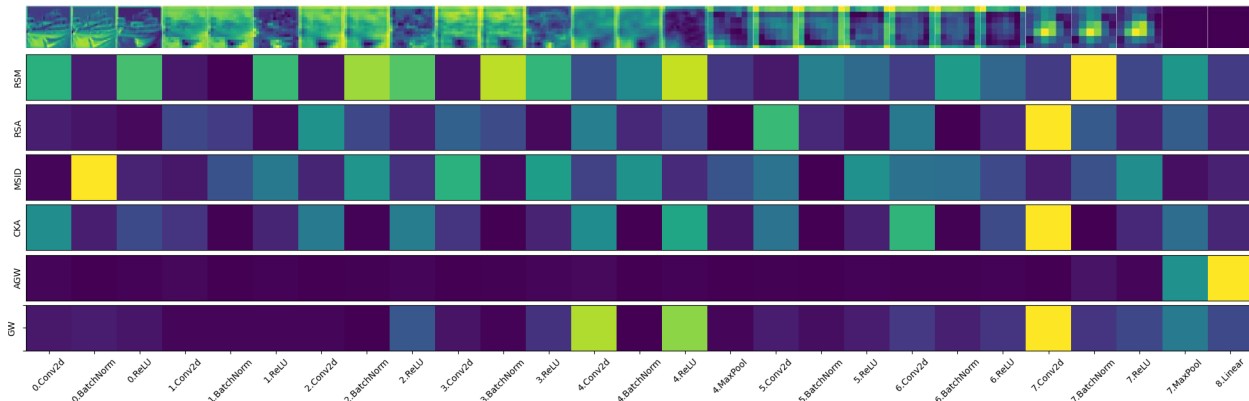

Figure 8: Pairwise layer distance between every layer and its previous layer on CIFAR-10, for various baselines, including RSM, RSA, MSID, CKA, AGW, and GW. Each Row Represents one method.

To further examine how the sub-network structures align with learned representation, we visualize the computed distances alongside the learned representations of an image of class ""ship" across all layers in Figure 8. The top row shows the representations of a ship at each layer. To see the gradual changes over layers, we visualize the distance between every layer and its previous layer, using various methods capable of handling different dimensions between compared spaces. Overall, RSM, RSA, MSID, and CKA show indicate significant changes across many layers, without clear evidence of sub-network structures. AGW highlights the changes in the final few layers only. In comparison, GW distance demonstrates the most consistency with the image representations visually. Specifically, the 3rd convolution layer (Layer ID 2.ReLU) introduces the first notable differences, where the ship's shape becomes less distinct, signaling the learning of mid-level features. The shapes become increasingly blurred in the 5th convolution layers (Layer ID 4.Conv2d) and by Layer 4.ReLU the ship's shape is nearly absent. The final convolutional layer (Layer ID 7.Conv2d) shows significant changes from its preceding layer (Layer ID 6.ReLU), marking the point where class-specific information is consolidated. These results suggest that GW distance aligns most effectively with the learned representations, providing strong evidence that it reveals meaningful layer structures.

## 6 Discussion

We proposed to model interpretation based on representation similarity within intermediate layers of neural networks, using GW distance to compute such similarities. To the best of our knowledge, our application of GW distance in this context is novel. On algebraic, real NLP, and vision tasks, we identified the existence of major groups amongst layers, corresponding to functionally meaningful abstractions. These results reveal implicit layer structure within neural networks, and highlight the potential sudden transition in network computation instead of smooth function change. Overall, our method provides an automatic approach to reveal layer grouping within neural networks, facilitating human understanding.

There are several limitations to our approach. GW is computationally more expensive than cosine/Euclidean and existing approximate solvers may introduce variance. Moreover, there are cases when invariance properties (such as permutation invariance) might mask meaningful structural differences, GW may not fail to reveal these changes. Future work could investigate other models and applications to observe general trends. Theoretical study of special properties of GW distances within the context of neural network interpretability is also an interesting future direction.

**Limitations and Broader Impact**  Our approach first assumes we have access to the intermediate layer representations, which may not be available for some black-box models. Our approach is general, but assumes the proposed distances correctly represent the representation similarities. Our findings are also limited to the datasets and models studied and are not guaranteed to be exactly the same in other scenarios. In terms of broader impact, our approach could be applied widely given its simplicity for identifying layer grouping in neural networks. However, more investigations on inner mechanisms will have to be done, perhaps building on our approach, in order to fully understand the behavior of neural models.

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

## A  Representations in Transformer-based and Convolution Neural Network

We consider multiple candidate $Y$'s to form the search space for target $Y^0$. In the context of MLP neural networks for example, where $\sigma(.)$ denotes the non-linearity and $W$s are the parameter matrices, we have $Y^* = W_n(\sigma(W_{n-1} \ldots \sigma(W_1 X)))$ for the whole network. We can extract many $Y$'s from intermediate functions of the model, for instance $Y_1 = W_1 X$, $Y_2 = \sigma(W_1 X)$, and so on. These $Y$'s are often called representations, activations, or sometimes even "outputs" from each layer.

For attention modules in transformer neural networks (Vaswani et al., 2017), we can similarly extract $Y$'s from attention key, query, and value functions as well as MLP functions. More specifically, a deep transformer architecture of depth $l$ is formed by sequentially stacking $l$ transformer blocks. Each transformer block takes the representations of a sequence $\boldsymbol{X}_{\text{in}} \in \mathbb{R}^{T \times d}$, where $\boldsymbol{X}_{\text{in}} = \text{Emb}(\boldsymbol{X})$ with embedding layer Emb and input $\boldsymbol{X}$, $T$ is the number of tokens and $d$ is the embedding dimension, and outputs $\boldsymbol{X}_{\text{out}}$, where:

$$\boldsymbol{X}_{\text{out}} = \alpha_{\text{FF}} \hat{\boldsymbol{X}} + \beta_{\text{FF}} \text{MLP}(\text{Norm}(\hat{\boldsymbol{X}}))$$
$$\text{where,} \quad \text{MLP}(\boldsymbol{X}_m) = \sigma(\boldsymbol{X}_m \boldsymbol{W}^1)\boldsymbol{W}^2$$
$$\hat{\boldsymbol{X}} = \alpha_{\text{SA}} \boldsymbol{X}_{\text{in}} + \beta_{\text{SA}} \text{MHA}(\text{Norm}(\boldsymbol{X}_{\text{in}})),$$
$$\text{MHA}(\boldsymbol{X}) = [\text{Attn}_1(\boldsymbol{X}), \ldots, \text{Attn}_H(\boldsymbol{X})]\boldsymbol{W}^P, \tag{2}$$
$$\text{Attn}(\boldsymbol{X}) = \boldsymbol{A}(\boldsymbol{X})\boldsymbol{X}\boldsymbol{W}^V,$$
$$\boldsymbol{A}(\boldsymbol{X}) = \text{softmax}\left(\frac{1}{\sqrt{d_k}}\boldsymbol{X}\boldsymbol{W}^Q \boldsymbol{W}^{K\top}\boldsymbol{X}^\top + \boldsymbol{M}\right),$$

with scalar weights $\alpha_{\text{FF}}$, $\beta_{\text{FF}}$, $\alpha_{\text{SA}}$, and $\beta_{\text{SA}}$ usually set to 1 by default. Here FF stands for feedforward network, SA stands for self-attention, MHA is Multi-Head Attention, and Norm is a normalization layer. MLP usually has a single hidden layer with dimension $d$ and ReLU activation. The MHA sub-block shares information among tokens by using self-attention with $\boldsymbol{W}^Q$, $\boldsymbol{W}^K$ and $\boldsymbol{W}^V$ indicating query, key and value matrices. We list the exact locations of representations considered in the transformer models in Table 4.

Table 4: Representations $Y$ in the attention-based model considered in experiments as per equation 2. Omitting $Y$ in most names for readability.

| (Across Blocks) | | | | | |
|---|---|---|---|---|---|
| Name | Resid-Pre$^l$ | $Y^l$, at each block | | | |
| Value | $= \boldsymbol{X}_{\text{in}}^l$ | $= \boldsymbol{X}_{\text{out}}^l$ | | | |
| (Within Each Block $l$) | | | | | |
| Name | Attn-Out$^l$ | Resid-Mid$^l$ | Pre | Post | MLP-out$^l$ | Resid-Post$^l$ |
| Value | $= \text{MHA}(\boldsymbol{X})^l$ | $= \hat{\boldsymbol{X}}$ | $= \hat{\boldsymbol{X}}\boldsymbol{W}^1$ | $= \text{MLP}(\hat{\boldsymbol{X}})$ | $= \text{MLP}(\hat{\boldsymbol{X}})$ | $= \boldsymbol{X}_{\text{out}}$ |
| (Within Each Attention Head $h$) | | | | | |
| Name | $k_h$ | $q_h$ | Attn-Pre$_h$ | Attn$_h$ | v$_h$ | z$_h$ |
| Value | $= \boldsymbol{X}\boldsymbol{W}^K$ | $= \boldsymbol{X}\boldsymbol{W}^Q$ | $= q_h k_h^T$ | $= \boldsymbol{A}(\boldsymbol{X})$ | $= \boldsymbol{X}\boldsymbol{W}^V$ | $= \text{Attn}(\boldsymbol{X})$ |

We also consider convolution neural networks for computer vision datasets. Specifically, we use a relatively lightweight ResNet9 (He et al., 2016; Park et al., 2023). The exact locations of the candidate representations considered are listed in Table 5.

## B  Modular Sum Experiment Details

We use the same architecture and protocols in training, as previous modular papers (Nanda et al., 2023; Zhong et al., 2024), based on their available Github repos. Specifically, we use transformer width $d = 128$, and each attention head has 32 dimensions. As a result, MLP has 512 hidden neurons. ReLU is used as the activation throughout the models,

Table 5: All representations $Y$ considered in ResNet 9 in experiments.

| (Module 0) | | | |
|---|---|---|---|
| Name | 0.Conv2d | 0.BatchNorm | 0.ReLU |
| Details | in-channel = 3, out =64, kernel size = (3,3) | Batch Normalization | activation |

| (Module 1) | | | |
|---|---|---|---|
| Name | 1.Conv2d | 1.BatchNorm | 1.ReLU |
| Details | in-channel = 64, out =128, kernel size = (5,5) | Batch Normalization | activation |

| (Module 2 & 3: Residual Block ) | | | |
|---|---|---|---|
| Name | 2.Conv2d | 2.BatchNorm | 2.ReLU |
| Name | 3.Conv2d | 3.BatchNorm | 3.ReLU |
| Details | in-channel = 128, out =128, kernel size = (3,3) | Batch Normalization | activation |

| (Module 4) | | | | |
|---|---|---|---|---|
| Name | 4.Conv2d | 4.BatchNorm | 4.ReLU | 4. MaxPool |
| Details | in-channel = 128, out =256, kernel size = (3,3) | Batch Normalization | activation | Kernel (2,2) |

| (Module 5 & 6: Residual Block ) | | | |
|---|---|---|---|
| Name | 5.Conv2d | 5.BatchNorm | 5.ReLU |
| Name | 6.Conv2d | 6.BatchNorm | 6.ReLU |
| Details | in-channel = 256, out =256, kernel size = (3,3) | Batch Normalization | |

| (Module 7) | | | | |
|---|---|---|---|---|
| Name | 7.Conv2d | 7.BatchNorm | 7.ReLU | 7. MaxPool |
| Details | in-channel = 256, out =128, kernel size = (3,3) | Batch Normalization | activation | Adaptive |

| (Module 8) | |
|---|---|
| Name | 8.Linear (classification) |
| Details | in-feature = 128, out =10 |

**Data**  Among all data points ($59^2 = 3481$ of them), we randomly select 80% as training samples and 20% as validation samples.

**Hyperparameters**  We used AdamW optimizer (Loshchilov & Hutter, 2017) with learning rate $\gamma = 0.001$ and weight decay factor $\beta = 2$. We use the shuffled data as one batch in every epoch. We train models from scratch and train for 26,000 epoches.

**Search Space**  For the $f_{\text{mod3}}$ dataset, we consider all layers in the network, including all representations within transformer blocks. As shown in Table 4, each attention head has 6 intermediate layers, for a total of 24. Each block has an additional 7 layers (1 input layer, Resid-Pre, and 6 intermediate layers). Hence, for three blocks each with four attention heads, we have a total of 93 representations to evaluate, as each block has $31 = 24 + 7$ representations.

**Training procedure**  We train 3 different neural networks with transformer blocks to predict $c$ given $(a, b)$. These networks contain input embeddings for $a$ and $b$, each of size $d$, i.e., $[\boldsymbol{E}_a, \boldsymbol{E}_b] \in \mathbb{R}^{2d}$, and predict a categorical output $c$ via an unembedding/decoding layer. All parameters in the network are learned. For the first simpler $f_{\text{mod}}$ dataset, we train a neural network consisting of a one-block ReLU transformer (Vaswani et al., 2017), following the same protocol and hyperparameter choices as previous works (Nanda et al., 2023; Zhong et al., 2024). We call this **Model 0**. For the more complex $f_{\text{mod3}}$ dataset, we train two neural networks consisting of three-block ReLU transformers, with 3 transformer blocks corresponding to the three levels of modular sum functions, and 4 attention heads within each block. The first network, which we call **Model E**, employs an end-to-end training procedure to directly learn output $c$ given input $(a, b)$. For the second network, which we call **Model L**, we use the same architecture as Model E but with a layer-wise training approach instead of end-to-end training. Specifically, we use the following 3-step procedure:

1) We train the first transformer block of Model L to predict $(c_1, b)$ using an additional linear layer on top, given inputs $(a, b)$. 2) Once block 1 is fully trained, we discard the linear layer, freeze everything before the linear layer, and use its representations of $(c_1, b)$ to train the second block to predict $(c_2, b)$, again

incorporating an additional layer on top. 3) Finally, we repeat the above step by freezing the first and second block and training the last block to predict $c$, using representations of $(c_2, b)$.

In all these models, we are able to achieve 100% prediction accuracy on a separate validation dataset.

To evaluate the capability of handling different dimensions, we directly measure GW distance between the 93 intermediate representation $Y$ (see appendix B for search space details) and $c$'s. To speed up computation of GW distance, we randomly sub-sample 1000 data from a total of 3600 samples, reducing time from 2 min to 5 seconds for each computation.

## C   Probes on Modular Sum Dataset: When Target is Known

When the target is a value from a known function, we can directly compare outputs between representations from each layer and the known function output. Representations from each layer can be directly compared with the target via a probe. We first consider **Model E** and then **Model L**.

**Linear Probe**   Popular linear probes can be used to assess the similarity between a target and any layer's representation. We perform linear regression of each target $(c_1, c_2, c)$ on each of the 93 representations $Y$, and report the residual error as the scoring distance function between $Y$ and $c$'s.

Table 6: Linear and Nonlinear Probe Results, for $f_{\mathrm{mod3}}$ dataset.

| Model L | Linear Probe for | Perfect Match? | Top Similar Layers | $D_{\min} =$ |
|---|---|---|---|---|
| | $c_1$ | ✓ | Resid-Post$^1$ and 21 others | 0 |
| | $c_2$ | ✓ | Resid-Post$^2$ and 21 others | 0 |
| | $c$ | ✓ | Resid-Post$^3$ and Post$^2$ | 0 |
| Model E | Linear Probe for | Perfect Match? | Top Similar Layers | $D_{\min} =$ |
| | $c_1$ | × | Post$^2$ | 0.522 |
| | $c_2$ | × | Post$^1$ | 0.93 |
| | $c$ | ✓ | Resid-Post$^3$ and 5 others | 0 |
| Model E | Nonlinear Probe for | Perfect Match? | Top Similar Layers | $D_{\min} =$ |
| | $c_1$ | ✓ | Resid-Post$^1$ and 15 others | 0 |
| | $c_2$ | ✓ | Resid-Post$^1$ and 4 others | 0 |
| | $c$ | ✓ | Resid-Post$^3$ and 9 others | 0 |

**Results**   Since we perform layer-wise training with **Model L**, we know the true locations of $c_1$ and $c_2$, which sit at $\boldsymbol{X}_{\mathrm{out}}^1$ and $\boldsymbol{X}_{\mathrm{out}}^2$ with names Resid-Post$^1$ and Resid-Post$^2$, respectively. As shown in the top part of Table 6, a linear regression probe can predict targets perfectly with these two layers. In fact, there are 21 other layers which also show perfect accuracy. For $c_1$, these consist of Post$^0$ and MLP-out$^0$ from the same block and some layers from the next block, including linear operations with all $k$'s, $q$'s, $v$'s. The final prediction $c$ can be linearly predicted as expected, due to the model's perfect prediction accuracy.

Naturally we would like to confirm if the same happens with **Model E**: if we use the same linear probe, does each block in **Model E** learn the corresponding $c$ at the output of the transformer block? As shown in the mid part of Table 6, we are not able to find any layer that produces a representation that is linearly predictive of $c_1$ and $c_2$, with the lowest prediction errors at 52% and 93%, respectively. Moreover, the most similar layers to $c_1$ and $c_2$ are in the 2nd block and 1st block respectively, instead of the expected 1st and 2nd blocks. This seems to suggest that **Model E** does not actually learn any function of $c_1$ and $c_2$.

**Non-linear Probe**   As discussed previously, to deal with the potentially large search space of functions of the target, a more powerful probe (such as a nonlinear MLP function) may have to be used so that it can detect more complex similarities to $c$. Therefore, we train a two-layer MLP[2] to predict $c$'s. As shown at the bottom of Table 6, these two-layer MLPs have more predictive power and can perfectly predict the

---

[2]We use the neural network classifier from the scikit-learn package, with default parameters.

targets, while still showing differences among various layers indicating that the matched layers do capture the intended target functions while other layers do not. Many layers in the 3rd block, for example, have only 1% accuracy relative to $c_1$. This indicates that non-linear probes can be used to find subgroups of layers in neural networks. Unlike existing work that primarily focuses on linear probes, we show that non-linear probes, still with limited capacity, are useful.

One issue with using predictive probes to compute the distance measure $D$ is that the target function has to be known. In practice, however, we may not know any intermediate targets, as suggested in the end-to-end training of **Model E**. While we still can try different target functions and use non-linear probes, the infinite number of possible targets makes such an approach inefficient. This calls for a different strategy to differentiate sub-components in a network through representation similarity.

## D    Real NLP Experiment Details

We analyze a BERT-base-uncased (Devlin et al., 2019) model based on our optimal matching inspired mechanistic interpretability approach. We fine tune it on two well known datasets in NLP; i) Yelp reviews (`https://www.kaggle.com/code/suzanaiacob/sentiment-analysis-of-the-yelp-reviews-data`) and ii) Stanford Sentiment Treebank-v2 (SST2), which is part of the GLUE NLP benchmark (Wang et al., 2019). Both of these are sentiment analysis tasks, where the goal is to predict if a piece of text has positive or negative sentiment. The Yelp dataset has hundreds of thousands of reviews, while the SST2 dataset has tens of thousands of sentences. The training details are as follows: i) Hardware: 1 A100 Nvidia GPU and 1 intel CPU, ii) Max. Sequence Length : 256, iii) Epochs: 1, iv) Batch Size: 16 and v) Learning Rate: $2e^{-5}$ with no weight decay. The accuracy on Yelp was 97.87%, while that on SST2 was 92.4%. Without fine tuning the pre-trained BERT models accuracy on Yelp and SST2 was 49.29% and 50.34% respectively indicative of random chance performance.

We also fine tuned a series of sparse models on these datasets. The method we used to sparsify was a state-of-the-art dynamic sparse training approach NeuroPrune (Dhurandhar et al., 2024), which leads to high performing structured sparse models. Using this approach and the same training settings as above we created BERT models with 25%, 70% and 95% sparsity which had accuracies of 96.31%, 97.53% and 96.22% respectively for the Yelp dataset and accuracies of 90.25%, 88.5% and 84.4% respectively for the SST2 dataset. We then used the resultant models for our analysis.

## E    GW Justification and Alignment

**Distance Distributions.** As an illustrative example, we plot the histogram on pairwise distances for a batch of samples across all transformer blocks in BERT models from the YELP review dataset in Figure 2. The results in Figure 2 show the distributions on pairwise distances begin to differ from block 9, consistent with GW distance observed in Figure 5, suggesting that significant transformations occur and can be effectively captured by GW.

**Neighborhood Change.** Complementary to the distribution of pairwise distances, the changing representations of samples could also alter their relative neighborhoods across transformer blocks. We plot a tSNE projection (Van der Maaten & Hinton, 2008) of representations from a batch of samples on YELP, and visualize it in Figure 2b and Figure 9e. The Jaccard similarity, measuring the overlap between top-5-neighbors of 3 selected samples across different transformer blocks, ranges from 0.0 to 0.43, with average values of $\{0.27, 0.26, 0.26\}$. The full details are shown in Table 7, as discussed below. Hence, the sample neighborhood changes across blocks, which can be indicative of functional changes that are not captured by comparing distributions alone. However, GW can account for such changes as well.

We plot a tSNE projection (Van der Maaten & Hinton, 2008) down to 2 dimensions, on a batch of 16 samples (color indicative of sample) on YELP, and visualize it in Figure 9e. As one can see, the sample neighborhood changes across layers, which can be indicative of functional changes but something that is not captured by comparing distributions. However, GW can also account for such changes.

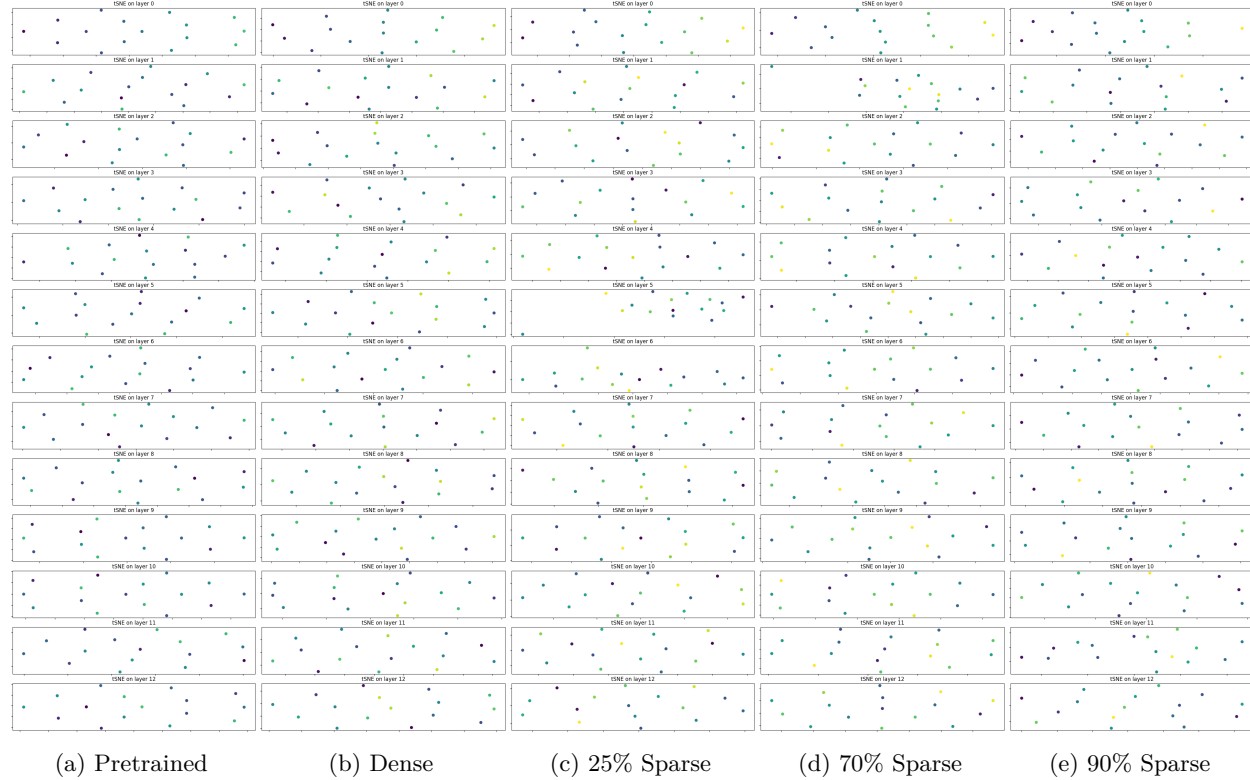

|                |                |                |                |                |
|:--------------:|:--------------:|:--------------:|:--------------:|:--------------:|
| (a) Pretrained | (b) Dense | (c) 25% Sparse | (d) 70% Sparse | (e) 90% Sparse |

Figure 9: tSNE projection on intermediate representations on Yelp, across BERT models with different sparsity levels. Different Rows: Results from all 12 transformer blocks, from top to bottom. Different columns: first column is the pre-trained BERT and the rest are fine tuned BERT models with increasing sparsity (dense, 25%, 70% and 95% sparsity).

Table 7: Jaccard Similarity on top-5-neighbors of Selected Samples across all transformer blocks.

| Sample 1 Block 0 v.s. | 1 | 2 | 3 | 4 | 5 | 6 | 7 | 8 | 9 | 10 | 11 | 12 | Mean |
|---|---|---|---|---|---|---|---|---|---|---|---|---|---|
| | 0.25 | 0.25 | 0.25 | 0.11 | 0.43 | 0.11 | 0.25 | 0.25 | 0.11 | 0.11 | 0.25 | 0.25 | **0.27** |
| Sample 2 Block 0 v.s. | 1 | 2 | 3 | 4 | 5 | 6 | 7 | 8 | 9 | 10 | 11 | 12 | Mean |
| | 0.11 | 043 | 0.11 | 0.11 | 0.11 | 0.0 | 0.11 | 0.25 | 0.25 | 0.43 | 0.25 | 0.25 | **0.26** |
| Sample 3 Block 0 v.s. | 1 | 2 | 3 | 4 | 5 | 6 | 7 | 8 | 9 | 10 | 11 | 12 | Mean |
| | 0.0 | 0.11 | 0.43 | 0.25 | 0.25 | 0.25 | 0.11 | 0.66 | 0.11 | 0.11 | 0.11 | 0.0 | **0.26** |

We also show Jaccard similarity measure on top-5-neighbors, per Euclidean distances on tSNE projection, of each of 3 samples across different transformer blocks. Jaccard similarity is a measure of two sets, computed as their intersection divided by their union. Results are shown in Table 7. This further shows the sample neighborhood changes across layers, and representation similarity measures should account for such changes.

To show the exact transportation plan from GW distances, we choose plot one batch of data with size 16, and show the transportation plan over 5 random layer pairs in Figure 10. As one can see, the transportation plan does not conform to identity-mapping. Both Wasserstein and Euclidean distance will likely have trouble handle in this case. We also note that the transportation plan shown Figure 10 is a permutation of the original data, rather than a distributed transportation plan. This behavior is consistent with existing Wasserstein optimal transport plan under certain conditions (Peyré et al., 2019).

To complement Figure 2 on other fine-tuned BERT models on YELP, we also plot all the histograms of pairwise distances between two samples in a batch, across all layers for each of 5 models in Figure 11. Pre-trained models are publicly available models training on other datasets. Row b) to e) are the fine-tuned models on YELP, with different sparsity levels. As one can see, pretrained models do not have much differentiations across layers in the histograms, with maximal KL-divergence of 0.11 between histogram in consecutive layers.

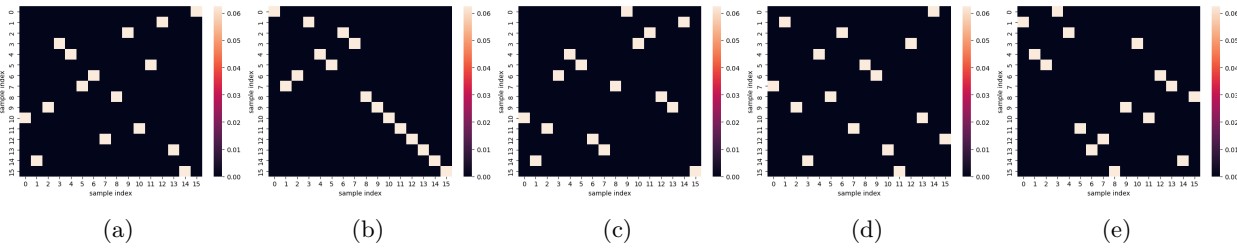

(a)          (b)          (c)          (d)          (e)

Figure 10: Pairwise GW transportation plan on Yelp, across BERT models. 5 of randomly chosen layer pairs are shown.

Fine tuned models, on the other hard, show larger KL-divergence values, in particular in later layers. For example, Layers 9 in the Dense BERT model contains KL distance of 1.58 from its previous layer. The results show that significant transformations in pairwise distances occur across layers and such distances would be captured by GW distances, as show in Figure 5 and Figure 13.

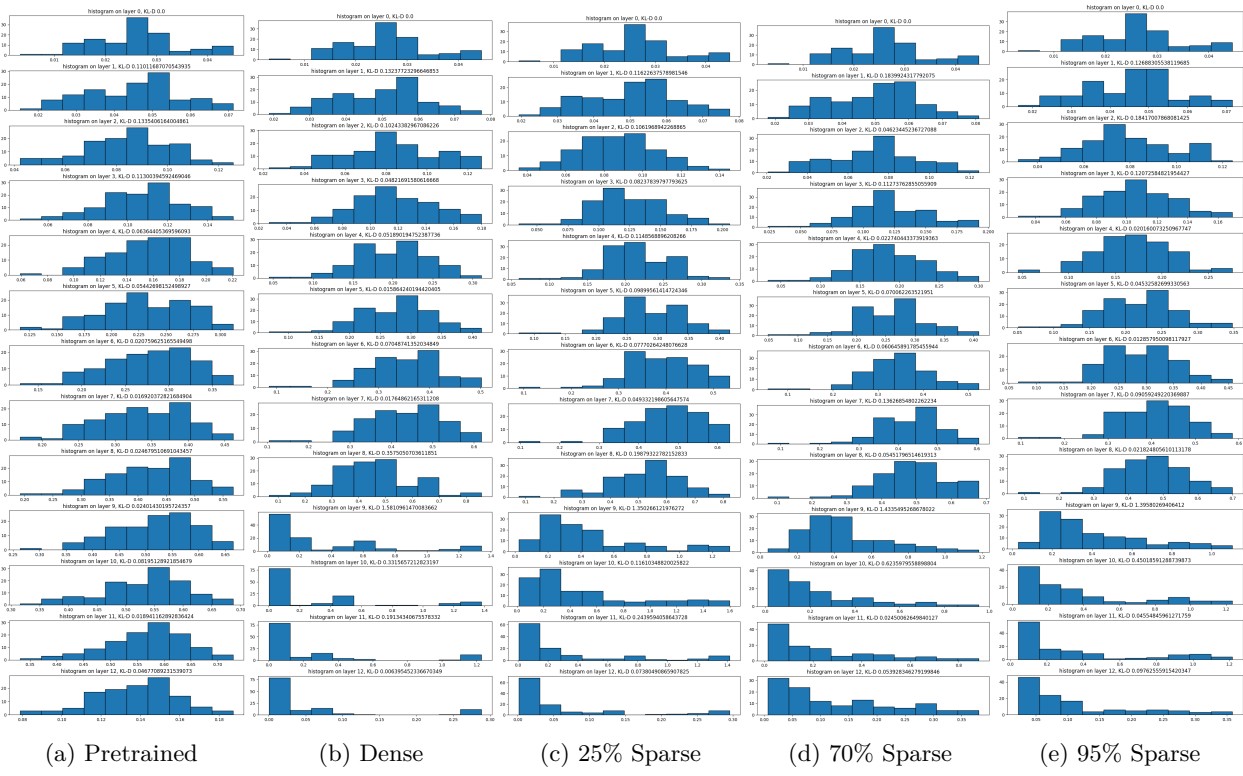

(a) Pretrained      (b) Dense      (c) 25% Sparse      (d) 70% Sparse      (e) 95% Sparse

Figure 11: Histogram on pairwise distances on Yelp, across BERT models with different sparsity levels. *a*) is the pre-trained BERT and the rest are fine tuned BERT models with increasing sparsity levels: (*b*)densely fine-tuned, *c*) 25%, *d*) 70% and *e*) 95% sparsity.

## F  Baseline Methods and Implementation Details

Besides the standard Euclidean, mutual information (MI) and cosine distances, we compare a few other baselines, as discussed below.

Wasserstein Distance (Dwivedi & Roig, 2019): We use the POT, python optimal transport library `pythonot` (Flamary et al., 2021), with the algorithm proposed in (Bonneel et al., 2011).

Representational similarity metric (RSM) (Klabunde et al., 2023): RSM compares two different spaces by using the $L_2$ norms on differences in inter-instances distances. This can be seen as approximation to GW using the fixed and identity transportation plan (i.e., the samples map to itself). We use existing implementation at: `https://github.com/mklabunde/llm_repsim/blob/main/llmcomp/measures/rsm_norm_difference.py`.

Representational Similarity Analysis (RSA) (Klabunde et al., 2023): RSA is similar to RSM but use correlation instead of $L_2$-norm to compute the final distance. Implementation at: `https://github.com/mklabunde/llm_repsim/blob/main/llmcomp/measures/rsa.py`

Canonical Correlation analysis (CCA) (Morcos et al., 2018): CCA compute distances based on variances and covariances. Implementation at: `https://github.com/google/svcca/blob/master/cca_core.py`

Centered Kernel Alignment (CKA) (Kornblith et al., 2019): CKA is based on normalized Hilbert-Schmidt Independence Criterion (HSIC). Implementation at: `https://github.com/mklabunde/llm_repsim/blob/main/llmcomp/measures/cka.py`

Multi-Scale Intrinsic Distance (MSID) Tsitsulin et al. (2019): MSID compute the intrinsic and multiple distance, and can be considered as a lower bound of the GW distance. Implementation at: `https://github.com/xgfs/imd/blob/master/msid/msid.py`. We have explored different hyperparameter settings with different neighbors k (5 or all batch data available) and number of iterations for SLQ, but results are all similar to the default parameter setting.

Augmented GW (AGW) (Demetci et al., 2023a): AGW considers feature alignment in addition to sample alignment. Its overall objective can be seen as a penalized GW distance. Implementation at: `https://github.com/pinardemetci/AGW-AISTATS24/tree/main`.

For all methods, we use default parameter settings to obtain results in the paper. Note that RSM, RSA, CCA, MSID, and AGW, along with our proposed approach can handle different dimensions of inputs.

Gromo-Wasserstein Distance (Dwivedi & Roig, 2019): We use the POT, python optimal transport library `pythonot` (Flamary et al., 2021). We use the solver based on the conditional gradient (Titouan et al., 2019).

## G   More Results on YELP

Due to the page limit, here we include baseline results on Yelp Datasets in Figure 12 and Figure 13.

**Setup** We now apply GW distance to real natural language processing tasks. We experiment on benchmark sentiment analysis datasets, Yelp reviews and Stanford Sentiment Treebank-v2 (SST2) from the GLUE NLP benchmark (Wang et al., 2019), with the goal to predict of the text has positive or negative sentiment, and analyze how different layers from fine-tuning BERT(-base) (Devlin et al., 2019) models perform on these datasets. We use the pretrained BERT to generate 4 fine-tuned models, corresponding to a dense model and 3 sparse models with sparsity levels of 25%, 70% and 95% using a state-of-the-art structured pruning algorithm (Dhurandhar et al., 2024). Sparsity are used to force models to condense information into the limited remaining weights, enabling us to examine potential links between this constraint and their structural similarity. Training details are in Appendix D. Due to the size of BERT models, we limit our analysis to comparing the final representations from each of the 12 transformer blocks, rather than examining all intermediate representations.

**Results** In Figure 5a, we see that the pre-trained BERT does not have major differences among blocks, which is not surprising given its accuracy on YELP is only 49.3% (roughly equivalent to random guessing). In Figures 5b to 5e, we see an interesting pattern emerge, revealing two-to-three major block structures in the (sparse) fine-tuned BERT models identified by our approach. The first major differences occur at block 9 and then the last three blocks (10, 11, 12) seem to form a distinct block. This seems to indicate that most of the function/task fitting occurs at these later blocks.

We compare the proposed GW distance with Euclidean, Cosine, and Wasserstein distance as baselines in Figure 12, on the same YELP dataset and with the same settings. Euclidean distance between two layers' outputs, shown in the first row of Figure 12, can be seen as the GW distance with a fixed identity-mapping

transportation plan for each sample. This validates the low-valued diagonal elements. Off-diagonal elements show greater variation, and it is less obvious there are two distinct sub-groups within layers. The similar pattern is also observed with Cosine and Wasserstein distances, with similar strong diagonal pattern but more pronounced block structures than Euclidean distance. we also include 6 other baseline similarity measure in Figure 13. Overall, CKA produces also similar block structures to the proposed GW distance, though with greater variability within block structures. In contrast, other baselines fail to reveal such clear block structures.

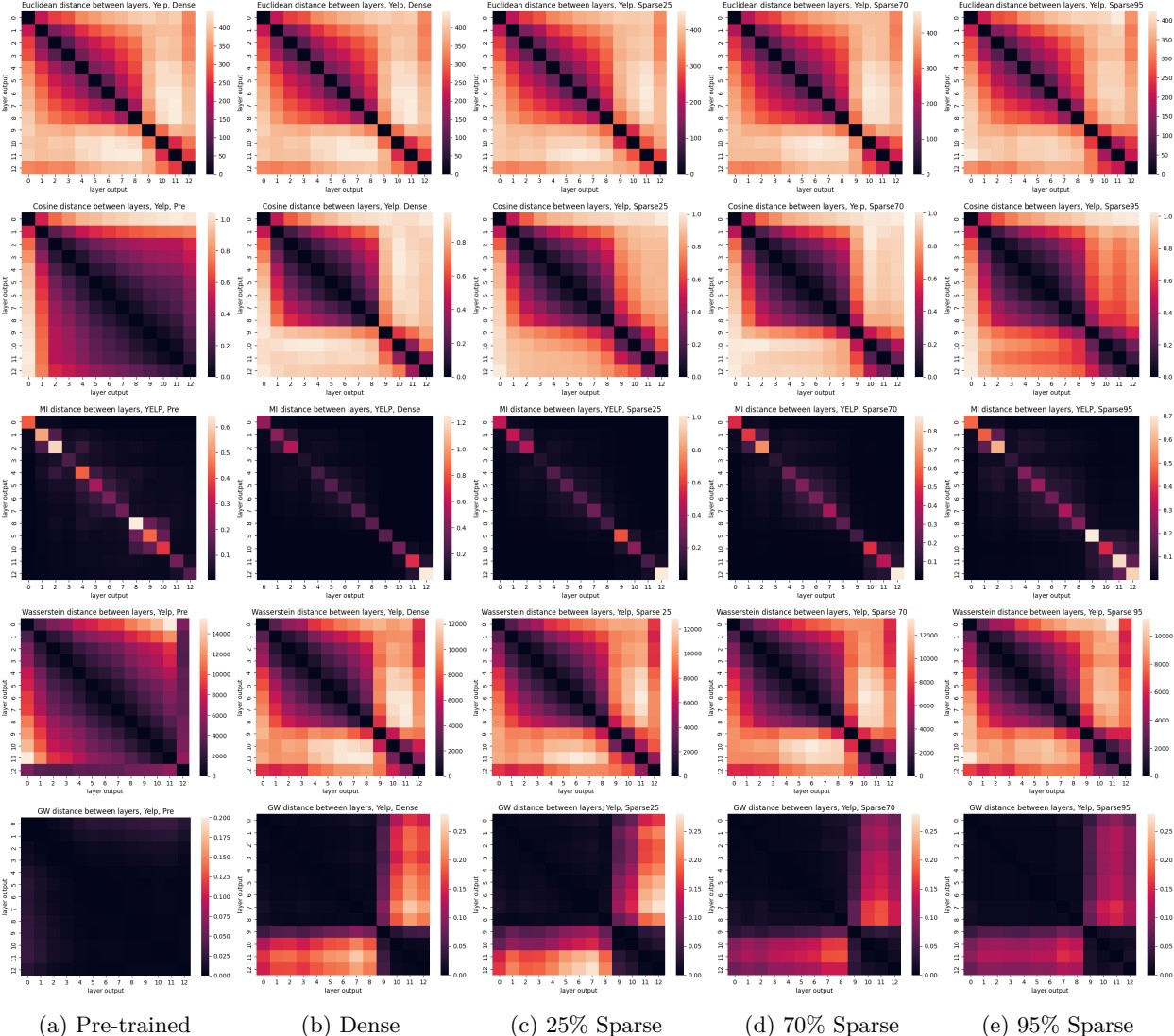

(a) Pre-trained  (b) Dense  (c) 25% Sparse  (d) 70% Sparse  (e) 95% Sparse

Figure 12: Pairwise (layer) distances on Yelp, across different BERT models. Different Rows: *Euclidean, Cosine, mutual information (MI), Wasserstein, and the proposed GW distance*, from top to bottom. Different columns: first column is the pre-trained BERT and the rest are fine tuned BERT models with increasing sparsity (dense, 25%, 70% and 95% sparsity). As can be seen GW clearly demarcates the (functional) sub-network blocks.

# H   Cross Model Comparison

We can also use GW distance to compare layers from different BERT models. Shown in Figure 14, pretrained and densely fine-tuned BERT models exhibit different similarity measures when compared to fine-tuned BERT models with different levels of sparsity.

## I  SST2 Datasets

Besides YELP Datasets, we also tested the GW distance on SST2 dataset. Results on SST2 dataset are shown in Figure 15 again confirm there exist two-three different groups in terms of functional similarity. The first major difference is seen at layers 10 and 11, while layer 12 forms its own block. When sparsifying these models, lesser differences are observed in general as also seen on the YELP dataset. Other baselines provide less clarity on the division of sub-components.

More baselines are included in Figure 16, as they do not all fit into the one page. Overall, RSA and CKA identify block structures but with larger 2nd block.

## J  GW Distance with Different Random Seeds

Neural networks initialized with different random seeds can converge to distinct representations (Li et al., 2015; Morcos et al., 2018; Kornblith et al., 2019), even when their performance is comparable. To study the impact of initialization seeds on the learned representations, we train the same BERT model on YELP datasets with different seeds, with identical hyperparameters for a total of 27,000 iterations. As shown in Figure 17, while the learned representations vary across seeds, but the general block structures remain consistent when analyzed using GW distances.

## K  Computer Vision Application: CIFAR-10 Datasets

In addition to the attention-based architectures, we also test our approach on ResNet9, a popular convolutional neural network architecture(He et al., 2016; Park et al., 2023). We compare a randomly initialized ResNet9 and a trained model on CIFAR 10 image dataset CIFAR-10 (Krizhevsky et al., 2009), achieving 91.63% accuracy on the test data. CIFAR-10 dataset consists of 60000 32x32 color images in 10 image classes, with 6000 images per class. There are 50000 training images and 10000 testing images. The classes are completely mutually exclusive. ResNet is a convolutional neural network with many residual connections. ResNet9 specifically contains 9 convolution layers, each followed by BatchNorm and ReLU activation. The exact details of the ResNet 9 is listed in Table 5.

We show the pairwise distance of all layers in consideration using all methods, that can handle difference dimensions of inputs, in Figure 18. The first column shows results from randomly initialized pre-trained models, and the second columns shows results from the trained ResNet. Pre-trained models generally do not show clear sub-network structures, while the trained models shows differences across layers. RSA, RSM, and CKA show progressive changes over the network layers, which is not too informative. AGW only shows the last a few layers contain significant changes, and MSID distance does not contain clear patterns. In comparison, GW distance shows clear division of 4 groups.

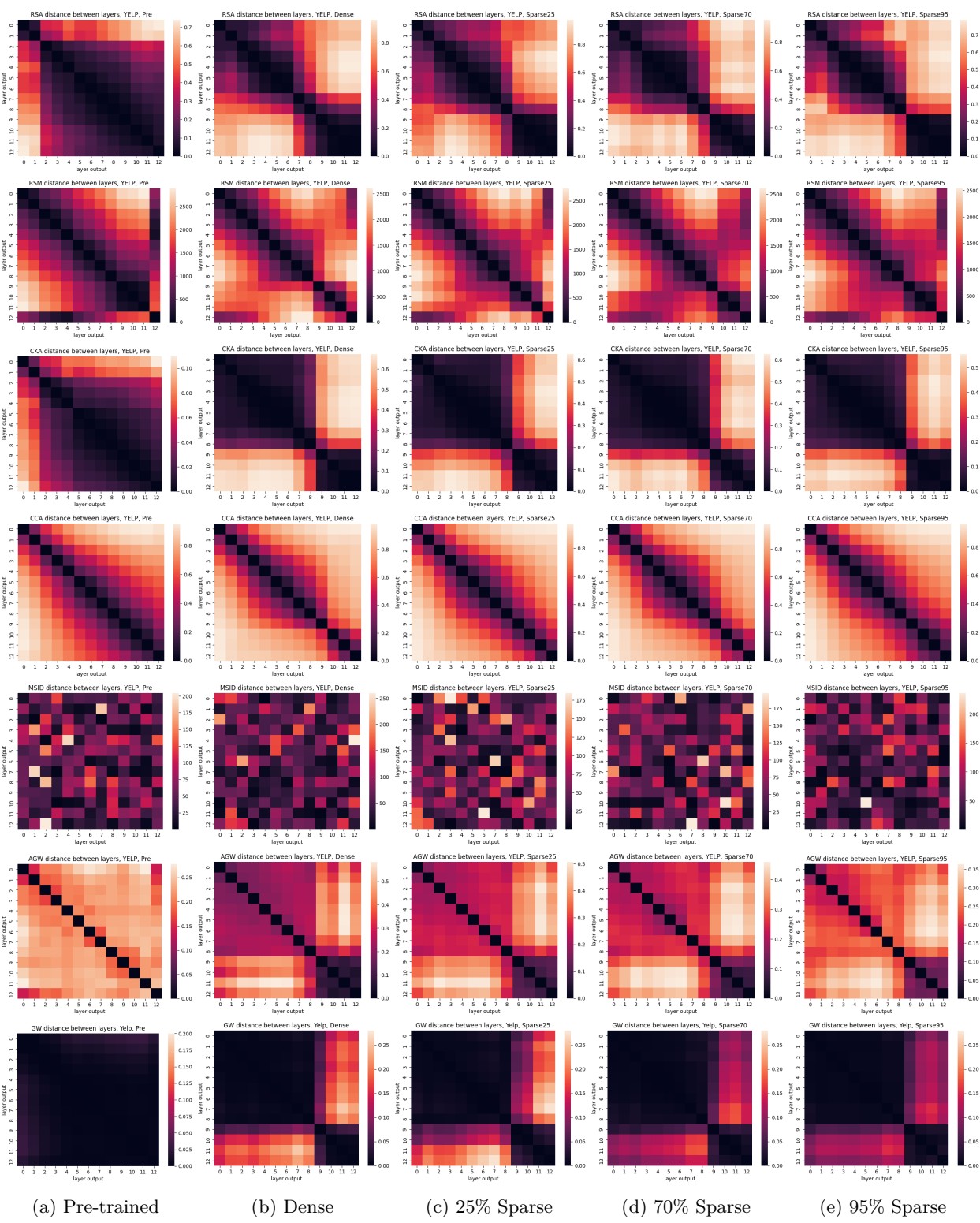

(a) Pre-trained  (b) Dense  (c) 25% Sparse  (d) 70% Sparse  (e) 95% Sparse

Figure 13: Pairwise (layer) distances on Yelp, across different BERT models. Different Rows: *RSA, RSM, CKA, CCA, MSID, AGW, and the proposed GW distance*, from top to bottom. Different columns: first column is the pre-trained BERT and the rest are fine tuned BERT models with increasing sparsity (dense, 25%, 70% and 95% sparsity). As one can be seen, GW clearly demarcates the (functional) sub-network blocks.

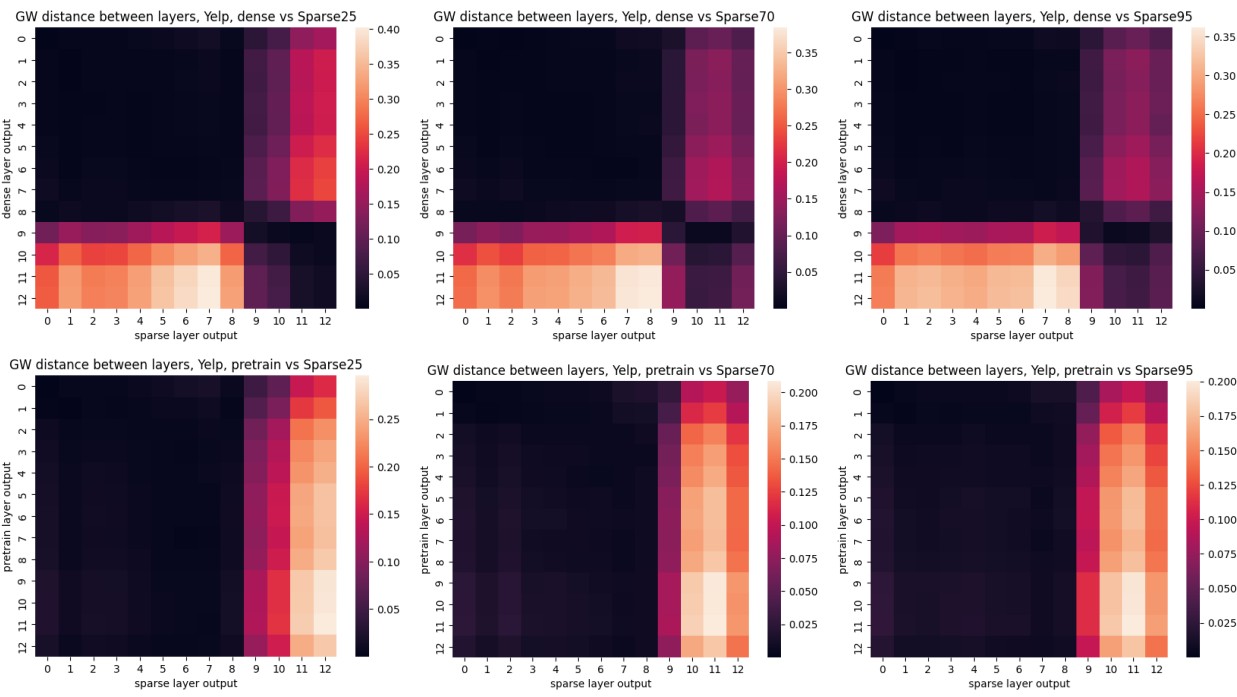

Figure 14: Pairwise distances on YELP dataset, of layers across two different BERT models. *TOP*: Densely fine-tuned BERT model vs fine-tuned BERT models with different sparsity levels. *Bottom*: Pretrained BERT model vs fine-tuned BERT models with different sparsity levels.

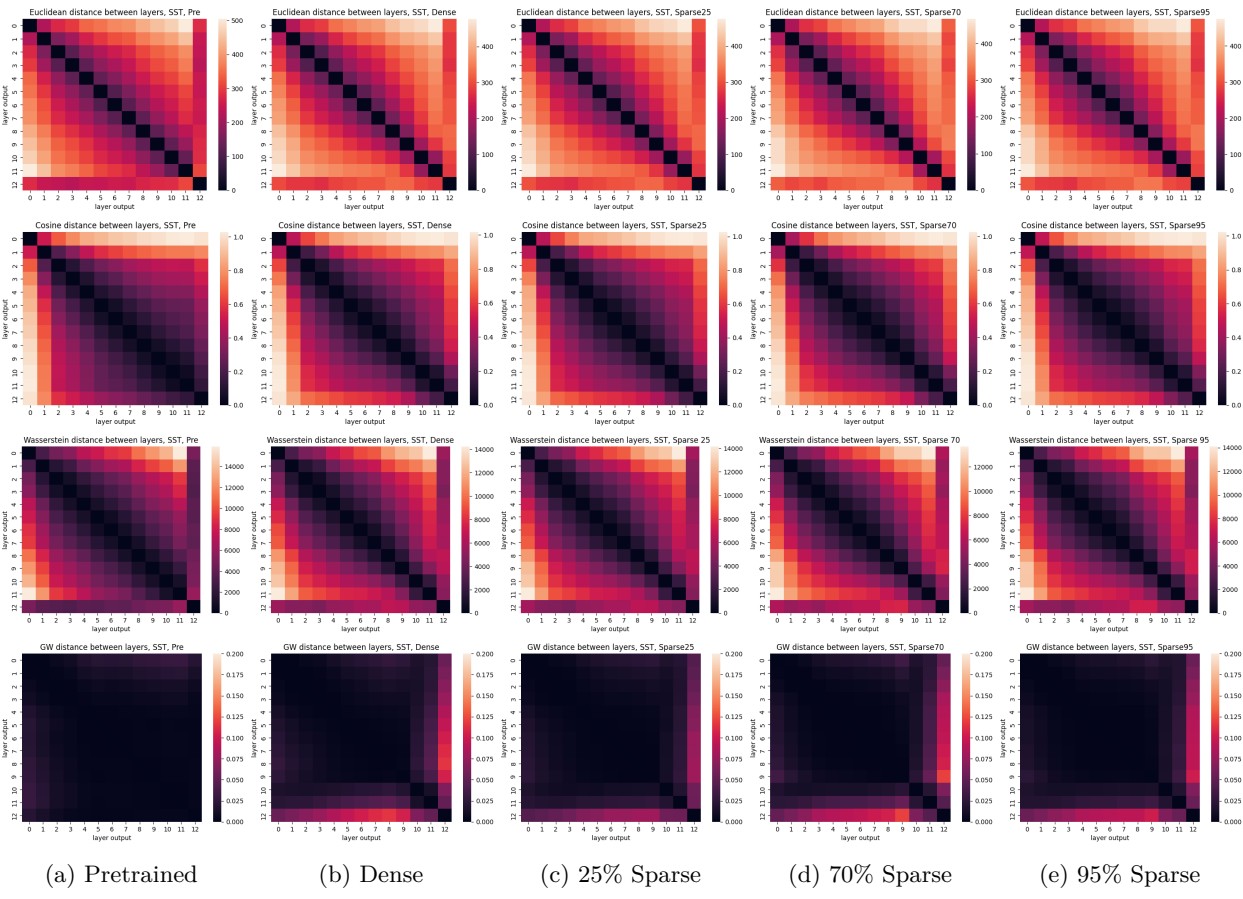

(a) Pretrained  (b) Dense  (c) 25% Sparse  (d) 70% Sparse  (e) 95% Sparse

Figure 15: Pairwise distances on SST dataset, across different BERT models. Different Rows: *Euclidean, Cosine, Wasserstein, and the proposed GW distances*, from top to bottom. Different columns: first column is the pre-trained BERT and the rest are fine tuned BERT models with increasing sparsity (dense, 25%, 70% and 95% sparsity).

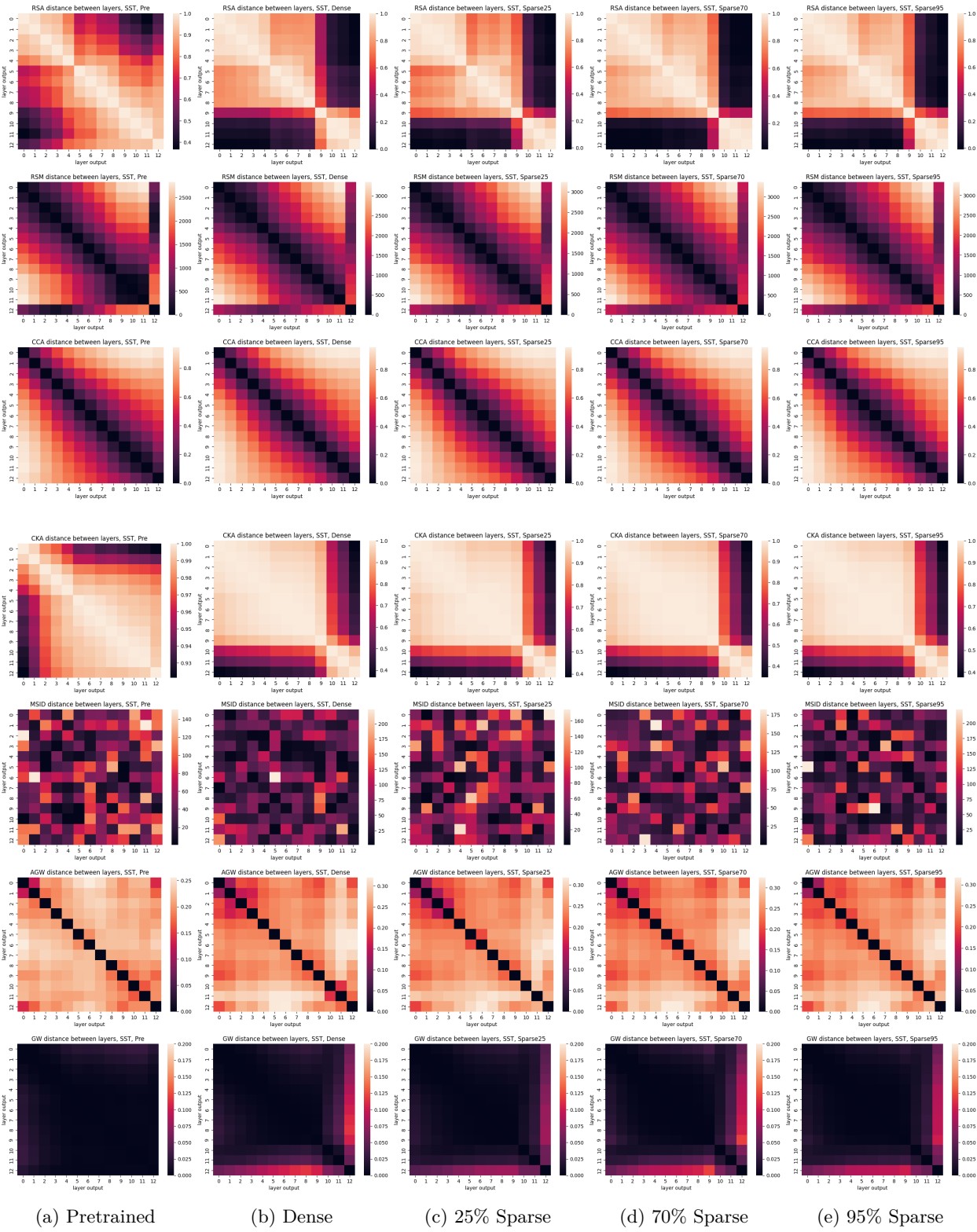

(a) Pretrained     (b) Dense     (c) 25% Sparse     (d) 70% Sparse     (e) 95% Sparse

Figure 16: More Pairwise distances on SST dataset, across different BERT models. Different Rows: *RSA, RSM, CCA, CKA, MSID, AGW, and the proposed GW distance*, from top to bottom. Different columns: first column is the pre-trained BERT and the rest are fine tuned BERT models with increasing sparsity (dense, 25%, 70% and 95% sparsity).

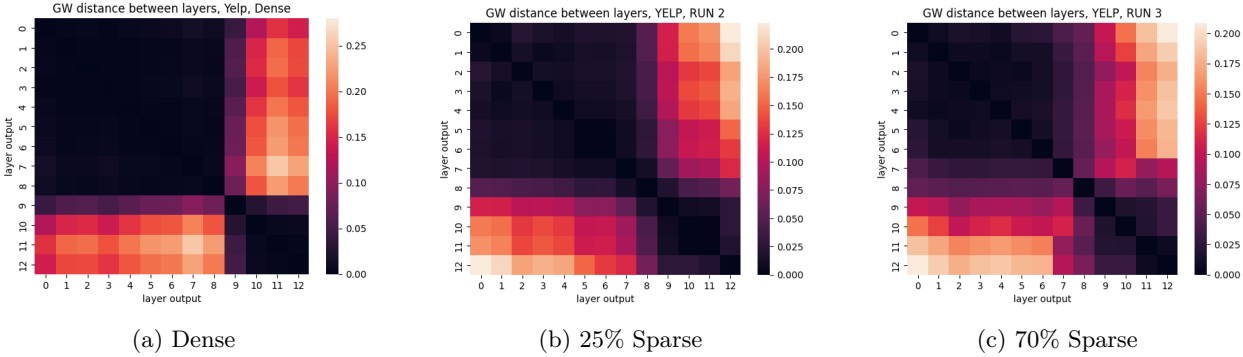

(a) Dense (b) 25% Sparse (c) 70% Sparse

Figure 17: Pairwise GW (layer) distances on Yelp, across BERT models trained with 3 different random seeds. As one can be seen, the (functional) sub-network blocks stay rather consistent with different seeds even though there is some variations among the models.

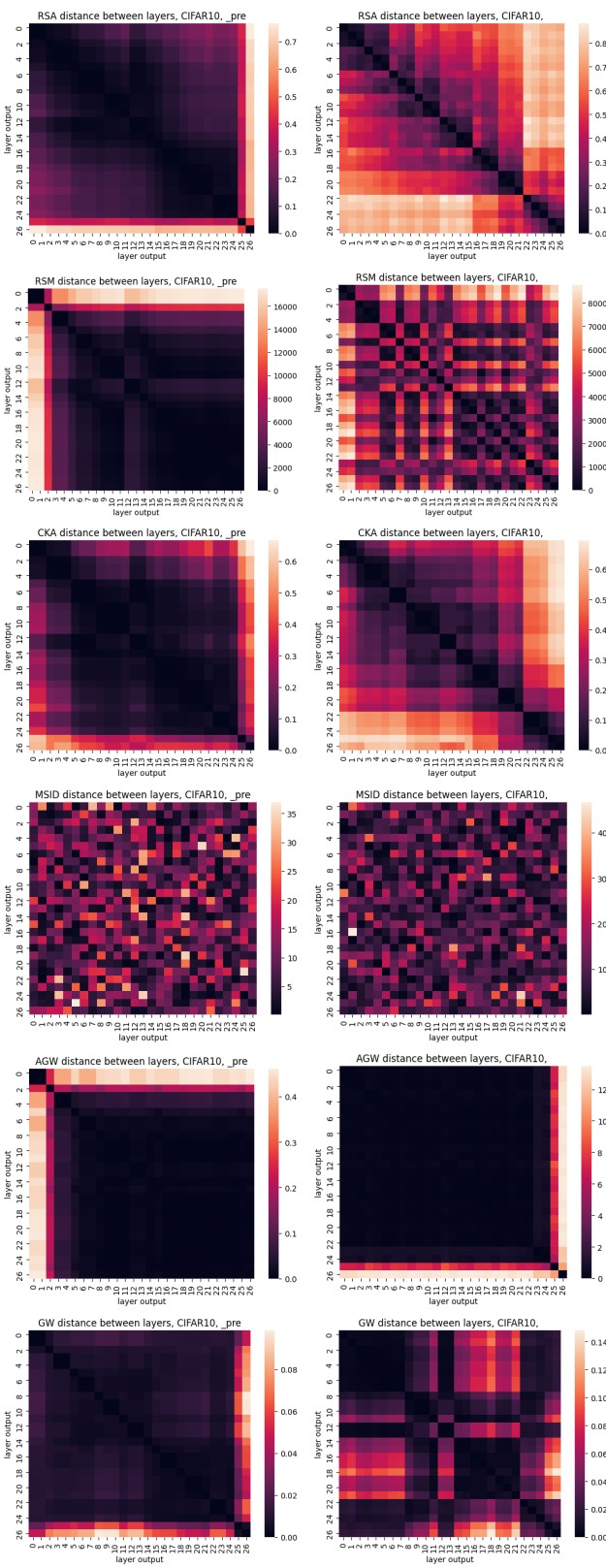

Figure 18: Pairwise (layer) distances on CIFAR-10, across different BERT models. Different Rows: *RSA, RSM, CKA, MSID, AGW, and the proposed GW distance*, from top to bottom. Different columns: the first column is the pre-trained ResNet9, and the 2nd column contains the fine tuned ResNet models.

