# OpenReview forum: "Representation Similarity Reveals Implicit Layer Grouping in Neural Networks"
_TMLR — Withdrawn by Authors_

### Review · Reviewer_TzUs · 2025-12-16

**Summary Of Contributions:**

The goal of this paper is to increase the understanding of where in neural networks some 'learning' emerges. This is done by finding groups of neural network layers, with similar output distributions, and hence at the borders of the groups layers with distinct output distributions. Such a grouping helps to understand where decisions are made in the network. The difficulty is which measure of similarity to use between layer outputs, Euclidean or cosine similarity measures have disadvantages (like requiring the same number of dimensions between layers). Instead the authors propose to use the Gromov-Wasserstein distance as an alternative. Having chosen this metric, the authors experimentally show pairwise layer distances on different tasks (BERT style networks trained on YELP, ResNet9 trained on CIFAR). Most of the experiments remain qualitative in nature, comparing pairwise distances of the proposed GW distance to existing methods, but without quantitative results indicating a better or more insightful metric.

**Audience:**

Yes

**Audience Explanation:**

Yes. There is definitely interested in deepening the understanding of neural network workings.

**Broader Impact Concerns:**

I foresee no broader impact concerns.

**Claims And Evidence:**

No

**Claims Explanation:**

This is the main question to address, and according to the TMLR acceptance criteria: this implies assessing the technical soundness as well as the clarity of the narrative and arguments presented. Based on the details below, my current answer is: No.

## 1. Clarity of the narrative

### 1.a English language in general
Overall I find that the manuscript contains many (small) language mistakes, typos, inconsistencies and vague language.
Examples include - but are not limited to:
 - inconsistent naming of terms, eg mlp vs MLP: both are used throughout the manuscript
 - Double capitals, like: Moreover, Various
 - Replication of words (eg or or, Another another)
 - Verb conjugations: eg a function that map x -> maps, the problem consisting -> consists
 - Plural / singular errors (eg There are many other approach to --> approaches, Fig 6 show a few --> shows)
 - Typos eg epoches

Moreover, there is some vague / unclear language, like [again not a full list]
 - has become a disparate area: disparate from what?
 - we identify layer groupings ... based on their representation at each layer: Where does 'their' refer to?
 - more holistic understanding of neural layers, how is this work more holistic than the other approaches providing a pairwise layer-to-layer similarity metric?
 - We consider multiple candidate Y ’s to form the search space for comparison: In all experiments just a single set of Ys is considered, so considering multiple candidates is not clear.
 - Generally it is unclear if the proposed method 'finds' groups, or only reveals them, as in there is no clustering algorithm, the pairwise layer-to-layer distances are used to visually identify groups.
 - We proposed to model interpretation based on representation similarity within intermediate layers of neural networks, using GW distance to compute such similarities. This is a long and unclear sentence.

Combining the grammar errors with the vague//unclear language, I conclude that the quality of the manuscript is insufficient for acceptance.

### 1.b Section 2: Background and Related Work
This is a dense section with many references to related work, but in general it fails to explain to what extent the mentioned papers are different or similar or are inspirations for the current work.

 - "Instead of studying circuits and neurons, we investigate differences among neural network layers as a whole" --> This seems to contradict Sect 4, the graph used each neuron (dimension) in a layer as a vertex in the Graph, so the distance is (partly) based on neuron-to-neuron similarities.
 - "we investigate differences among neural network layers as a whole, based on an existing line of work (Nanda et al., 2023)." --> The method of Nanda is not discussed, the reference is mostly used to refer to their dataset / setup.
 - Related, block structures within neural network layers have been observed in previous studies (Nguyen et al., 2022). --> This reference is not used beyond this sentence. How is the current method different, what different kinds of blocks can be found by the selected (GW) distance metric?
 - The main contribution of this paper is a different layer-to-layer pairwise distance, then please compare it mathematical and conceptually to the most obvious competitors (including  RSM (Klabunde et al., 2023), RSA (Klabunde et al.,2023), CCA (Morcos et al., 2018), CKA (Kornblith et al., 2019), MSID (Tsitsulin et al. 2019), Wasserstein (Dwivedi & Roig, 2019), and AGW (Demetci et al., 2023a). Including a table with the desired (and missing) properties and maybe the core functions.

I conclude that the embedding of the current manuscript in related work is insufficient.


## 2. Technical Soundness

The paper does not introduce any new technical / mathematical formulation, it proposes to use the existing GW distance for a layer-to-layer distance, and in Sect 4.1 it successfully justifies why the GW distance could be used for this layer-to-layer analysis. So, in that sense it is a sound contribution.

However, the technical description in Sect 3 (Representation Similarity) and Sect 4 (GW Distance) is *not* sufficiently detailed to ensure the technical soundness or to reproduce the (conceptual) methods proposed in the paper:
- Sect 3 is way too long and verbose for saying just three things: (a) A neural network is a composition of Networks; (b) we can study the output after layer i and j; (c) X is a matrix of n inputs with dimension d_x and Y is a matrix of n outputs with dimension d_y.
- Sect 3 the search space key argument feels unnatural, every layer is used as an output. Do the pairwise similarities change if more layers are added? How does the resulting pairwise diagram look if other choices are made? It seems as simple as to use all layers, maybe make a choice using the layer output before or after an activation function.
- Sect 4 Here you lost me, an illustration might be a good addition.
 a. First there is a move from layer-to-layer similarities to similarity within a layer - where each output neuron is a vertex in a graph, and weights between vertices are based on the 'distance between two neurons', however the manuscript does not state which distance measure is used.
 b. This results in a N x N distance function for each layer: D1 and D2 in Eq 1. The manuscript does not detail (a) how the graph is used to define a distance of a single example, nor which distance metric is used between sample i and j.
 c. The discrete distributions over the representations (μ1 and μ2) are not defined.
 d. Why is the matching between 'the n examples', while the goal is 'best permutation of representation dimensions in one layer that aligns with vertices in another layer'.


This lack of clarity / explanation at a conceptual level and in the details in the most relevant sections of the manuscript makes that I rate the technical soundness as insufficient.

## 3. Experimental evidence.

In general the pairwise layer-to-layer similarities mostly serve visual inspection and hence qualitative results. These are mostly performed on (only) three scenarios: (a) fine-tuning a pre-trained BERT model on the YELP dataset; (b) some synthetic modular sum task; and (c) ResNet-9 trained on CIFAR-10. While I appreciate the diversity, these network designs are rather old, and shallow compared to deep networks used for most vision and language tasks nowadays.

Remaining remarks:
- Figure 2a: What is the KL measuring? What is in the histogram? There are 16 samples? - Figure 2b: Is this a projection of 16 samples in a tSNE embedding? What should be visible from here? tSNE plots can show almost anything depending on the settings, these look like the samples are rather uniform?
- This makes the neighborhood change 'experiment' also rather anecdotal instead of rigorous.
- Appendix J & Figure 17: The experiment is to study the influence of random seeds on the block structure of the found layer groups. It is unclear what the Figure 17 is showing, the subtitles says it relates to sparsity.
- Figure 5: If the layer block structure is task dependent (as suggested in the text), what is then the influence on the layer structure of another task using the same pre-trained network?
- Figure 5: Since groupings are created by inspection of the pairwise distances, many of the other methods also show a grouping with two layer-groups (e.g. cosine, RSA, or AGW).
- Figure 8: It feels rather anecdotal to draw conclusions upon visual inspection of a single image of a single category.

Regarding the claim, GW pairwise similarity can help to find layer groupings, I rate the current experimental evidence as borderline / reject. While indeed the experiments do indicate that GW could be an interesting measure, I'm missing larger / deeper network architectures, different tasks to show different groupings in similar architectures, and beyond anecdotal evidence.

**Requested Changes:**

I request the following changes:
- [Critical] Overall improvement of the language / flow of the manuscript.
- [Critical] Improvement of the embedding into related work.
- [Critical] Improvement of the technical section with enough details to reproduce the method and experiments.
- [Critical] Improvement of the description of the experiments and the results.
- [Strengthening] Include more / deeper networks in the analysis.
- [Strengthening] Include a study on which (sub)layers to use in the analysis.
- [Strengthening] Include a study of multiple tasks on the same network architecture to see if layer groupings are task or network-design specific.

---

### Review · Reviewer_s7jp · 2025-12-18

**Summary Of Contributions:**

The manuscript aims to provide means to improve the efficiency of mechanistic interpretation of neural networks. To achieve this, this work proposes to use Gromov–Wasserstein (GW) distance to compare representations across layers. The authors argue that GW distance is particularly suitable for this task since it can be used as a similarity measure even when the layers to be analyzed in neural networks are of different dimensionalities.

The proposed GW distance to generate pairwise layer-distance matrices to visually (and via clustering) identify “implicit layer groups”, whose representations are similar, separated by “transition” boundaries, where representations can change more abruptly.

Empirically, the work visualizes clear block-diagonal patterns on (i) synthetic algebraic tasks, (ii) BERT fine-tuned on YELP / SST2, and (iii) a ResNet9 on CIFAR-10; and argues that the discovered boundaries can guide partial fine-tuning (freezing early blocks) and pruning/compression with minimal/low accuracy drop (only done for BERT).

**Additional Comments:**

No comments.

**Audience:**

Yes

**Audience Explanation:**

I think mechanistic interpretability is of interest to the audience of TMLR. I would even argue that there is an even increased interest for this topic in the ML research community.

**Broader Impact Concerns:**

No concerns.

**Claims And Evidence:**

No

**Claims Explanation:**

As far as I can see, the manuscript makes the following claims (page 2, bottom):

- **(C1)**: GW distance provides a systematic and architecture-agnostic way to analyze and identify layer structure

- **(C2)**: our findings provide a holistic view of differences in representations of models trained with different strategies

- **(C3)**: observe clear patterns in the form of block structures among different layers, suggesting there exist layer groups that potentially compute different functions, particularly at the transition layers where major functional changes may occur.

- **(C4)**: Moreover, our approach is applicable to various downstream tasks, such as tracking the emergence of layer groups during training process (§ 5.3) and identifying potentially redundant layers in model compression and fine-tuning (§ 5.2)

- **(C5)**:  our method can improve the efficiency of mechanistic interpretation by finding layer groups in any neural model, reducing the need for extensive human effort and contributing to a further understanding of neural network behaviors.

I could agree with the claim **(C1)**, **(C3)**, but I do not see the claim **(C2)**, **(C4)**, and **(C5)** being supported by the manuscript.

**(C2)**: I do not see experiments where different training strategies are used as foundation for GW's analysis.

**(C3)**: I do not see this for all experiments, and it is not compared to other baselines as seen in my statement below

**(C5)**: since most of the results are of qualitative nature, showing visualizations of layer similarities, I do not think that the proposed method "reduces the need for extensive human effort".

However, the major concern that I have about this work is that it mostly provides qualitative results, i.e., visualizations of layer similarities and compares them to other methods using other visualization of layer similarities. Since there is no ground truth telling us which visualization is the right one, one can only look at these plots and compare them to visualization of other methods showing very similar block structures (e.g., CKA, RSA).

For example, in Fig. 4, one can see that (a) RSA, (c) CKA, and (f) GW show very similar block structure. Similarly, in Fig. 5, one can see that (f) RSA and (j) GW do show similar block structure, and it is very difficult to argue that GW shows a better block structure than RSA. Maybe the plot of RSA is a more truthful representation of layer grouping than GW. However, I would not be able to argue *for* or *against* this by just looking at the visualizations.

The only experiment that provides quantitative results are the BERT experiments about "model compression" and "model pruning". Unfortunately, these experiments are only done on GW and not using the other baseline methods that are listed in the qualitative results. I could argue that maybe using the emerging RSA block structure, one could do a more compact model compression or more efficient model pruning than using GW's block structure.

**Requested Changes:**

In general, I think that the manuscript needs more quantitative results clearly demonstrating the proposed methods' utility as compared to other baselines. Please see my concerns as outlined above.

I would appreciate to improve Fig.1, as some depictions are confusing and some information is missing. I assume that the left part depicts a transformer block with self-attention being depicted as h_1, ..., h_3 and multiple blue arrows that are pointing into nothing and are not further explained. I also have difficulties to understand what the pre-attention layer, attention layer, and post attention layer mean or if they correspond one-to-one to the pairwise similarity matrix or not? Looking at the matrix, it would be interesting to know which layers correspond to which in the transformer block, and a legend would also be helpful.

---

### Review · Reviewer_VbZz · 2026-01-19

**Summary Of Contributions:**

This paper uses a novel plus several other representational similarity metrics to compare layers in a network amongst themselves and study if layer groups emerge.

**Audience:**

Yes

**Audience Explanation:**

I have read and reviewed similar papers in TMLR and I believe the broad readership as well as the subfield studying similarity measures would be interested in this paper.

**Broader Impact Concerns:**

A statement is present in the paper, and I believe an ethical concerns statement is not required in this paper. However, from what the authors wrote, it seems that they misunderstood what the broader impact statement entails and have instead talked about whether their approach can be deployed broadly.

**Claims And Evidence:**

No

**Claims Explanation:**

Some of the analyses seem reasonable, but I am not convinced about the authors' reasoning about whether a new metric was needed to show these results, or that the properties the authors ascribe to the GW metric are not also present in other metrics, or even that there is a universal agreement that the properties the authors describe as desirable properties of a metric are in fact always desirable.

Here are some objections I have with the authors' claims:

1) Wrong axioms: On page 2, when motivating the need for GW, the authors note that "Additionally, representations should be invariant to transformations such as rotation, scaling, permutation, and reflection". While desirable in many contexts, this is not a universally desirable property of a metric, for example, look at SoftMatching distance by Khosla and Williams where being sensitive to rotations is especially desirable. Moreover, the authors probably meant to write 'metrics' instead of 'representations' in the statement I quoted above, because representations in a conv layer, for example, would not be invariant to rotation.

2) Most of the analyses could be carried out if GW was never designed at all, unlike what the authors seem to claim. Although I always appreciate new metrics, the paper lacks a good justification for it. The authors talk about invariance (please see point 1 about my objection to that). The authors talk about usability for unequal dimensions, which again is not a problem with several of the other metrics used as a benchmark.

3) The authors make statements such as "GW distance gives the most distinctive layer groups." But in the absence of a ground truth about what the 'ground truth' grouping should be, we don't know if the grouping found by another metric is the grouping that reflects what is actually happening in the network. One way to solve this problem would be to take a model with known computations such as HMAX or a Bayesian Ideal Observer or a pyramid of known operators and use these metrics to see which metric would best be able to explain the grouping the actual known functions would predict. There is no strong quantitative justification about why one metric's predictions should be supported over another.


This paper is looking at layer grouping in networks using GW. Problems similar to grouping in networks have been studied before, for example in papers by Maithra Raghu. So the main contribution would be the GW metric (similar approaches to which also exist, which the authors themselves acknowledge.) But it doesn't seem like the paper has established that the metric is required or is better than the alternatives.

Other issues:

4) Table 2 should be compared against results if the freezing procedure was done based on groups found by other methods.


5) Why is CKA not present in Fig 5 and Fig 12? This is the only metric that would show a grouping structure similar to GW per the authors, so why is it not applied wherever applicable? Why is the metric choice not consistent between figures?


6) These seem like duplicate citations:

Pinar Demetci, Quang Huy Tran, Ievgen Redko, and Ritambhara Singh. Revisiting invariances and introducing
priors in gromov-wasserstein distances. arXiv preprint arXiv:2307.10093, 2023a.
Pinar Demetci, Quang Huy Tran, Ievgen Redko, and Ritambhara Singh. Revisiting invariances and introducing
priors in gromov-wasserstein distances. arXiv:2307.10093, 2023b.

7) RSA reference should be some paper from the Kriegeskorte lab instead of a review paper. If you wanted to cite a review paper, why are other metrics cited based on their papers but RSA based on a review paper? Klabunde et al talk about multiple of these metrics. (Note: I am not associated with the Kriegeskorte lab, as the editor can verify.)

8) Fig 2, Fig 9, Fig 11 - make them more readable

9) Typo, Page 3: "There are many other approach to understand neural network inner mechanism, such as weight inspection and manipulation.” Approach -> approaches.

10) “One possible explanation could be that Model E directly operates in the trigonometry space (Nanda et al., 2023), without having to predict the exact integer values until later, thereby suppressing the distances.” This seems like an easy hypothesis to test, based on previous approaches.

11) The authors mention " Looking at Model L we see predominantly 3 groupings of layers: i) layers roughly from 20 to 44 are similar to each other and to layers 52 to 72, ii) layers roughly 12 to 19 are similar to each other and layers 45 to 51 and iii) the initial and last few layers are mainly similar to themselves. Interestingly, the number of groupings corresponds to the 3 functions trained layer-wise in Model L. " Whether there is something deeper to this interesting correspondence can easily be checked by designing an f mod p task and seeing if p groups emerge now.

12) Page 10: “we study the problem of model compress or pruning”. Compress → compression.

13) Typo: MSIDTsitsulin et al. (2019) on Page 6.

14) Typo: Page 3:  Several similarity measures (Tsitsulin et al., 2019; Demetci et al., 2023a) seek different approximation or or addition to GW distance.  (Or or -> or).

**Requested Changes:**

Better justify the need for GW.

Analyze functions where the 'ground truth' grouping is known.

Address issues pointed out in points 1-14 above.

---

### Note · Authors · 2026-02-22

**Comment:**

We appreciate the reviewers’ time and constructive feedback, which will help us improve the work. We are withdrawing this submission based on the reviewers’ feedback to revise the work more thoroughly.

**Withdrawal Confirmation:**

I have read and agree with the venue's withdrawal policy on behalf of myself and my co-authors.